# Towards On-Policy SFT: Distribution Discriminant Theory and its Applications in LLM Training

**Miaosen Zhang** [* 1 2]  **Yishan Liu** [* 1 3]  **Shuxia Lin** [1]  **Qi Dai** [2]  **Chong Luo** [2]
**Baining Guo** [1 2]  **Weihao Jiang** [3]  **Peng Hou** [3]  **Anxiang Zeng** [3]  **Xu Yang** [1]  **Xin Geng** [1]

## Abstract

Supervised fine-tuning (SFT) is computationally efficient but often yields inferior generalization compared to reinforcement learning (RL). This gap is primarily driven by RL's use of on-policy data. We propose a framework to bridge this chasm by enabling On-Policy SFT. We first present ***Distribution Discriminant Theory (DDT)***, which explains and quantifies the alignment between data and the model-induced distribution. Leveraging DDT, we introduce two complementary techniques: (i) ***In-Distribution Finetuning (IDFT)***, a loss-level method to enhance generalization ability of SFT, and (ii) ***Hinted Decoding***, a data-level technique that can re-align the training corpus to the model's distribution. Extensive experiments demonstrate that our framework achieves generalization performance surpassing prominent offline RL algorithms, including DPO and SimPO, while maintaining the efficiency of an SFT pipeline. The proposed framework thus offers a practical alternative in domains where RL is infeasible. We open-source the code here: https://github.com/zhangmiaosen2000/Towards-On-Policy-SFT.

## 1. Introduction

Reinforcement learning (RL) (Sutton et al., 1998; Ouyang et al., 2022) and supervised fine-tuning (SFT) are two key methods in the post-training stage of large language models (LLMs) (Zhao et al., 2023; Zhu et al., 2025b; 2024b). RL often exhibits stronger generalization than SFT (Xu

et al., 2021; Lin et al., 2025), whether used to improve value alignment via RL from Human Feedback (RLHF) (Ouyang et al., 2022) like ChatGPT (Schulman et al., 2022), or to strengthen reasoning via RL from verifiable rewards (RLVR) (Lightman et al., 2023) such as DeepSeek-R1 (Guo et al., 2025). However, RL also has drawbacks. First, when obtaining reliable, verifiable feedback is difficult (e.g., resource-intensive agent settings (Yao et al., 2022), mathematical proof problems (Yang et al., 2023)), or biased (Gao et al., 2023), RL becomes hard to apply. Second, current RL for LLM typically provides only a terminal reward signal. This sparse supervision yields a much lower learning efficiency than SFT. For example, if repeated rollouts on a given instance fail to improve the reward, RL training tends to stall (Zelikman et al., 2022).

Therefore, an alternative path is to retain SFT's data efficiency while enhancing its generalization. A prevailing view in prior work is that the key distinction between SFT and RL lies in RL's use of on-policy data (Chu et al., 2025; Ouyang et al., 2022), which preserves the model's native distribution and mitigates catastrophic forgetting (Yuan et al., 2025; Liu et al.). In contrast, standard SFT forces the model to fit all external data equally. Enforcing the model to learn data with a large distribution gap can damage its pre-trained knowledge structures. Thus, distinguishing whether a sequence matches the model's internal distribution is essential for understanding the learning process and preventing catastrophic forgetting. This naturally raises our central question: Can we bridge the SFT-to-RL chasm by aligning the training process with the model's own distribution, both at the data level and at the objective level, while preserving the original knowledge?

In this work, we show that it is feasible. We first consider the problem of how to directly quantify what constitutes in-distribution data. Through systematic comparison, we identify the centered log-likelihood (CLL) (Cox, 1961) as the optimal criterion for distribution discrimination. Our contribution for the theory part is that we leverage signal detection theory (Macmillan, 2002) to provide a novel linear-threshold decision perspective and establish the optimality of CLL. Furthermore, in the sequential setting, we derive

[1]Department of Computer Science, Southeast University, Nanjing, China [2]Microsoft Research Asia, Beijing, China [3]Shopee, Shanghai, China. Correspondence to: Xu Yang <>, Xin Geng <>, Baining Guo <>.

*Proceedings of the 43rd International Conference on Machine Learning*, Seoul, South Korea. PMLR 306, 2026. Copyright 2026 by the author(s).

error bounds for using the token-summed CLL as an LLM distribution test. These results constitute our **Distribution Discriminant Theory (DDT)**, which provides a rigorous foundation grounded in actual LLM generative mechanics. Extensive experiments on recent advanced LLMs (Guo et al., 2025; Grattafiori et al., 2024; Yang et al., 2025), validate the theory and align with its predictions.

Building on DDT, we develop two direct applications to improve SFT. First, at the loss level, we reweight the SFT objective using our distributional criterion and introduce **I**n-**D**istribution **F**ine**T**uning (**IDFT**), allocating more weight to tokens that are in-distribution for the model, thereby preserving its native distribution and mitigating catastrophic forgetting. Second, we apply DDT into the decoding process for dataset re-alignment. We propose **Hinted Decoding**: given a question and its answer, it decodes a response aligned with the model's distribution.

Training a base model on a fixed dataset, under the same setup as prior work that enhances SFT generalization (Wu et al., 2025; Diao et al., 2026), our IDFT delivers substantial gains. By applying Hinted Decoding to rewrite the dataset into in-distribution samples and then training with IDFT, our approach surpasses offline RL methods (Rafailov et al., 2023; Yin et al., 2024; Xu et al., 2024; Meng et al., 2024)) on the same data while using less compute and achieves higher data efficiency. We hope this work empowers scenarios that are hard to apply RL, inspires further research.

**Related works.**  Several works explored the differences between SFT and RL. Theoretical analyses have characterized SFT as minimizing the forward KL divergence, which induces mode-covering behavior and forces the model to average over the data distribution, potentially limiting its ability to filter out noise (Chu et al., 2025; Kirk et al., 2023). In contrast, RLHF and methods like DPO approximate the reverse KL divergence, promoting mode-seeking behavior that concentrates probability mass on high-reward regions (Chu et al., 2025). Furthermore, prior work (Xu et al., 2021) analytically shows that RL methods achieve superior out-of-distribution generalization by optimizing policies on self-generated rollouts rather than fixed datasets. However, these works often lack a direct quantification of the distributional proximity. Our work contributes additional insights.

In terms of enhancing the generalization capability of SFT, recent works have proposed hybrid objectives (Liu et al., 2026a)to bridge this gap. Wu et al. (2025) proposed Dynamic Fine-Tuning (DFT), which dynamically reweights the SFT loss based on token probabilities to mitigate the high variance of implicit rewards. Building on this, Anchored Supervised Fine-Tuning (ASFT) (Zhu et al., 2025a) incorporates a trust-region constraint to prevent distributional drift, while other approaches like ProFit (Liu et al., 2026b) selec-

tively mask low-probability tokens to prevent surface-level overfitting. However, these approaches simply re-weight the SFT objective, often lacking an interpretation of data distribution perspective(Zhu et al., 2026; 2024a). Besides, our experiments demonstrate that when using stronger models (e.g., instruct models) where there is a significant gap in data distribution (Lin et al., 2025), simply modifying the SFT loss(Tan et al., 2026; Weng et al., 2026) is insufficient. Therefore, we also optimize at the data level and train with instruct models, comparing our results with offline RL.

## 2. Distribution Discriminant Theory

To realize the distributional alignment proposed in our central question, we first need to quantify what constitutes 'in-distribution' data for a given model. In this section, we establish the **Distribution Discriminant Theory (DDT)**, framing the detection of distribution alignment as a statistical hypothesis testing problem.

**Notations.**  Let $\mathcal{V}$ denote a finite vocabulary. We consider a sequence of tokens $\mathbf{x} = \{x_1, x_2, \ldots, x_n\}$, where each $x_t \in \mathcal{V}$. At each time step $t$, the language model induces a conditional probability distribution over the next token given the context $c_t = (Q, x_{<t})$, denoted as $p_t(\cdot) := p_\theta(\cdot \mid c_t)$, where $\theta$ is the parameter of LLM. Denote the standard Shannon entropy at step $t$ with context $c_t$ as:

$$H[p_t] := -\sum_{v \in \mathcal{V}} p_t(v) \log p_t(v) = \mathbb{E}_{X \sim p_t}[-\log p_t(X)].$$

In this work, we treat the detection of distribution alignment as a statistical decision problem. We introduce the following hypothesis testing framework:

**Assumption 2.1.** For any observed token $x_t$, we consider two mutually exclusive hypotheses:

- $\mathcal{H}_0$ (In-distribution): The token is sampled from the model's own distribution, $x_t \sim p_t$.
- $\mathcal{H}_1$ (Out-of-distribution): The token originates from an unknown external mechanism $x_t \sim q_t$, where $q_t \neq p_t$.

To evaluate the discriminative capability of a given statistic scorer $S$ in distinguishing $\mathcal{H}_0$ from $\mathcal{H}_1$, we employ the *Signal-to-Noise Ratio (SNR)*, a commonly used and reliable metric derived from *Signal Detection Theory* (SDT) (Macmillan, 2002):

$$\text{SNR}[S] := \frac{(\mathbb{E}[S \mid \mathcal{H}_1] - \mathbb{E}[S \mid \mathcal{H}_0])^2}{\text{Var}(S \mid \mathcal{H}_0)}.$$

We provide an intuitive explanation in Figure 1. Simply distinguishing the means of two classes for a given statistic is not sufficient. The SNR is negatively correlated with the overlap between the two distributions. The larger the SNR,

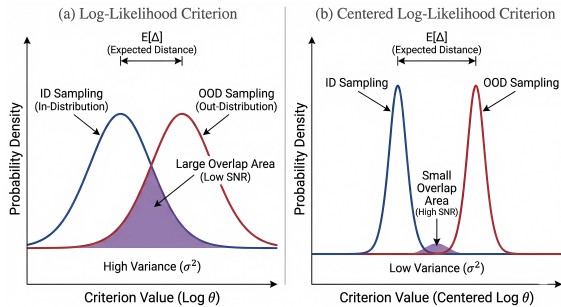

Figure 1. Intuitive understanding of SNR.

the better the distinction. We note that for the ideal case of normal distributions, the following conclusion from (Green et al., 1966) holds:

*Remark* 2.2. The overlap area between two normal distributions with equal variances is given by $S_{overlap} = 2 \cdot \Phi(-\sqrt{\text{SNR}/2})$ where $\Phi$ is the standard normal cumulative distribution function.

Based on the above information, this work proves the following content, all proofs can be found in Appendix A.

**Theorem 2.3.** *Given the context of using a statistic $S$ to distinguish $\mathcal{H}_0$ from $\mathcal{H}_1$, Consider the operator family $\mathcal{J} = \{\log p(x) + \mathcal{C}[p]|p \in \Omega, \mathcal{C} : \Omega \mapsto R\}$, where $p$ is a probability density and $\mathcal{C}$ is a real-valued functional on the space of densities $\Omega$ (e.g., entropy, mean). We have:*

$$argmax_{S \in \mathcal{J}} \quad SNR[S] = \log p(x) + H[p]$$

This result takes the form of the Centralized Log-Likelihood (CLL) criterion, while $\log p(x)$ is log-likelihood (LL). We also provide an explanation for the necessity of using $\log p(x) + \mathcal{C}$ family, rather than other families like $p(x) + \mathcal{C}$, in Appendix A.4.

**Corollary 2.4.** $SNR[S_{CLL}] \geq SNR[S_{LL}]$, *and when an LLM employs any decoding method other than greedy decoding, $SNR[S_{CLL}] > SNR[S_{LL}]$ holds strictly.*

**Definition 2.5** (SNR-Optimal Distribution Criterion). We define the SNR-optimal criterion for LLM at position $t$ as:

$$\varphi_t(\cdot) := \log p_t(\cdot) + H[p_t]. \tag{1}$$

For practical engineering applications, we propose a bounded version of this criterion: $\tilde{\varphi} = \text{clip}(\varphi, -B, B)$.

We also provide an intuitive explanation of why $\varphi$ can effectively determine whether data originates from the model's own distribution. As illustrated in Figure 3, when data is sampled from the model's distribution, it is highly likely to be sampled near the probability peaks. If these probability peaks are small, the probability distribution naturally flattens due to the requirement that the integral of the probability is one, resulting in increased entropy.

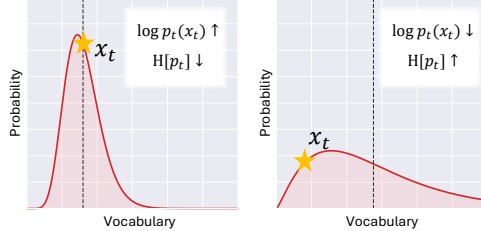

Figure 3. A simple illustration of why the criterion keeps stably high when data is sampled with model's distribution. The yellow star represents the temperature sampled token.

We also discuss the sequential statistical properties. Define the cumulative score trajectory for a sequence of length $L$ as $S_k = \sum_{t=1}^{k} \varphi_t$. We identify the stochastic behaviors of $S_k$ under the two hypotheses.

**Proposition 2.6** (Martingale Property under $\mathcal{H}_0$). *Under the in-distribution hypothesis $\mathcal{H}_0$, the sequence of cumulative scores $\{S_k\}_{k \geq 0}$ constitutes a zero-mean martingale with respect to the filtration $\mathcal{F}_{k-1}$ generated by the history. That is, for all $k \geq 1$:*

$$\mathbb{E}[S_k \mid \mathcal{F}_{k-1}, \mathcal{H}_0] = S_{k-1}.$$

**Proposition 2.7** (Negative Drift under $\mathcal{H}_1$). *Under the hypothesis $\mathcal{H}_1$, assuming the tokens are generated by $q_t$, the sequence $S_k$ exhibits a systematic linear negative drift determined by the Kullback-Leibler divergence:*

$$\mathbb{E}[\varphi_t \mid \mathcal{H}_1] = -D_{\text{KL}}(q_t \| p_t) \leq 0.$$

**Proposition 2.8** (Error Bound of $S_n$). *The probability that an in-domain sequence of length $L$ is misclassified (i.e., $\tilde{S}_L = \sum_{t=1}^{L} \tilde{\varphi}_t < -\gamma$) decays exponentially(Freedman, 1975):*

$$\mathbb{P}(\tilde{S}_L \leq -\gamma \mid \mathcal{H}_0) \leq \exp\left(-\frac{\gamma^2}{2(V_L + B\gamma/3)}\right),$$

*where $V_L$ is the cumulative conditional variance.*

**Empirical validation** We conducted validation experiments across multiple models and various types of datasets, as shown in Figure 2. For the given problems in the dataset, we recorded relevant metrics for the dataset's responses (in red) and the model's self-generated responses (in blue).

At the sample level, the $\tilde{S}_n$ metric effectively differentiates between in-distribution and out-of-distribution (OOD) data. By examining the distribution of $\varphi$ at the token level, we gained the following insights: whether a piece of data aligns with the model's distribution is not determined by the overall distribution of tokens, as depicted in Figure 2(b), but rather by the impact of a small number of outliers on the total $\tilde{S}_n$, as shown in Figure 2(c). This aligns with the principles of natural language: even if the language styles of two individuals or models differ greatly, the majority of token

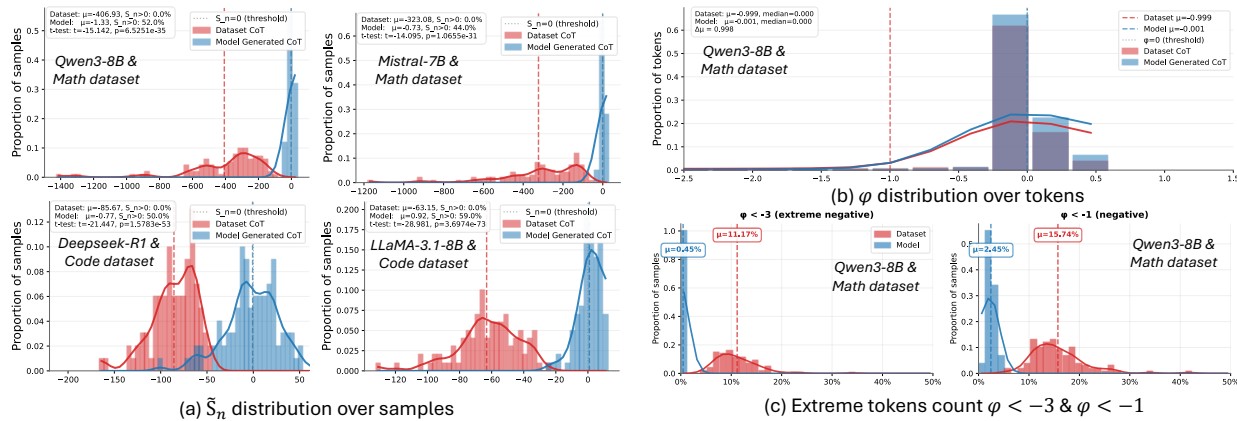

*Figure 2.* Empirical validation of the theory with multiple advanced LLMs and data types. More results in Appendix A.5.

usage is determined by the need to maintain grammatical coherence. Style differences manifest in critical elements such as conjunctions and transitional words, which involve only a small number of tokens.

## 3. Applications

### 3.1. In-Distribution Finetuning

Building upon the sequential properties, we propose **In-Distribution Finetuning (IDFT)** to address the inherent pathologies of standard SFT.

Standard SFT operates on the assumption that every token in the training set represents an absolute ground truth. Consequently, its objective ($\mathcal{L} = -\log p_t$) imposes severe penalties on prediction errors: as the probability $p_t \to 0$, the gradient magnitude $\propto 1/p_t$ explodes towards infinity. This indiscriminate penalization mechanism is perilous. When the training data contains noise (e.g., hallucinations, labeling errors) or samples significantly beyond the model's current capabilities, SFT induces the model to overfit these patterns via drastic parameter updates. This aggressive fitting disrupts the model's pre-trained general structures, serving as a primary cause of **catastrophic forgetting**. IDFT aims to mitigate this by introducing a self-aware regulation mechanism based on $\varphi_t$ in equation 2.5. As illustrated in Figure 4, while SFT blindly forces gradients on OOD tokens leading to instability, IDFT adaptively suppresses these harmful signals through dynamic modulation.

**Adaptive Mapping and Objective.** We construct a modulation coefficient $\gamma_t$ that dynamically adjusts the learning intensity based on the token's statistical status. The mapping is defined as:

$$\gamma_t(\varphi_t) = \exp(-\varphi_t). \qquad (2)$$

The exponential form is selected to provide a reciprocal response to the exponential sensitivity of the log-likelihood gradient. Unlike bounded saturating functions such as sigmoid, $\exp(-\varphi_t)$ offers a sufficient dynamic range to neu-

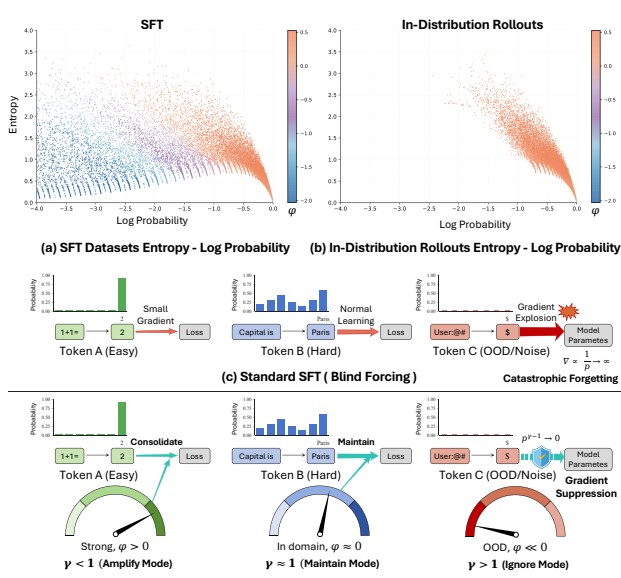

*Figure 4.* Visualization of the statistic of tokens of dataset and model's rollouts (a & b), inspired by (Diao et al., 2026). The illustration of the core effects of IDFT (c).

tralize the $1/p_t$ singularity while maintaining a smooth transition centered at the martingale equilibrium where $\varphi_t = 0$. This mapping continuously modulates the loss curvature: for out-of-distribution tokens ($\varphi_t \ll 0$), $\gamma_t > 1$, triggering gradient suppression; for in-domain tokens ($\varphi_t \approx 0$), $\gamma_t \approx 1$, recovering standard probability-weighted learning; and for strong-domain tokens ($\varphi_t > 0$), $\gamma_t < 1$, enhancing knowledge consolidation. The final IDFT objective is formally defined as the expectation over this adaptive Poly-Log family:

$$\mathcal{L}_{\text{IDFT}}(\theta) = -\frac{1}{L}\sum_{t=1}^{L} p_t(x_t)^{\gamma_t} \log p_t(x_t). \qquad (3)$$

**Gradient Dynamics and Controlled Learning.** Analytically, the gradient of the proposed loss scales approximately with $p_t^{\gamma_t - 1}$. This term acts as an adaptive gate: it vanishes for low-probability tokens when $\gamma_t > 1$ (suppression) and

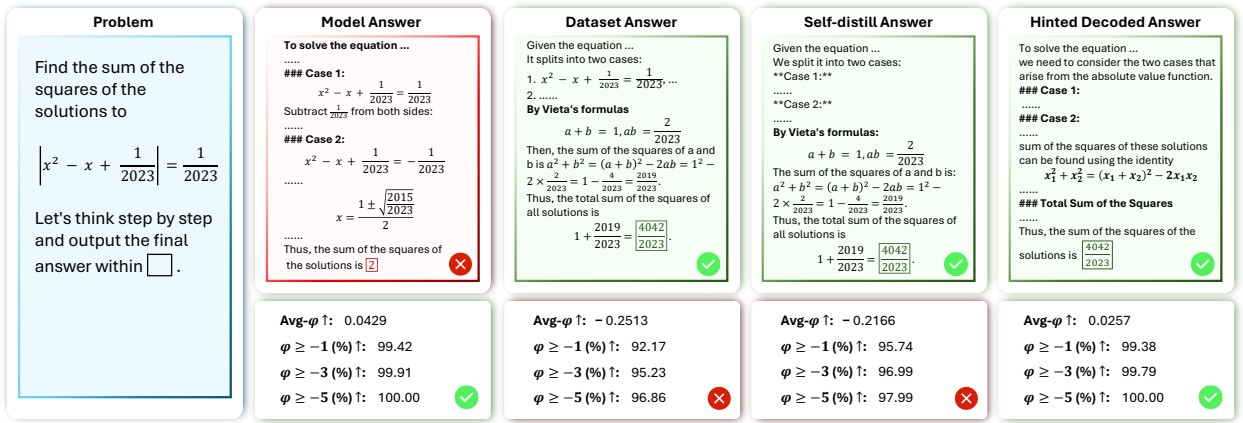

*Figure 5.* Comparison of different data. Question picked from Numina-Math. Hinted decoded response keeps the model's styling (e.g., markdown style) while remaining the correct answer. More case study can be found in Appendix D.2.

intensifies for high-confidence tokens when $\gamma_t < 1$ (reinforcement).

This mechanism enforces a **controlled learning schedule** tailored to the model's current capacity. By initially dampening statistically distant samples ($\varphi_t \ll 0$), IDFT protects existing structures from destabilizing gradients until the model adapts and $\varphi_t$ rises. While related to methods like DFT (Wu et al., 2025) that reinforce high-confidence tokens, IDFT differs by utilizing the theoretically optimal CLL statistic $\varphi_t$. Unlike approaches based on raw $p_t$, IDFT decouples intrinsic context difficulty from distributional alignment, ensuring that suppression is driven by true OOD shifts rather than task complexity. This theoretical grounding allows for a more precise preservation of pre-trained knowledge during the assimilation of new domains.

### 3.2. Hinted Decoding

Another application of DDT is the design of a decoding algorithm that transforms responses from general datasets into those that align with the model's distribution, specifically in a step-by-step chain-of-thought manner. Before introducing our algorithm, we first present a prompt engineering based, common and straightforward baseline approach:

**Self-distillation baseline.** (Shenfeld et al., 2026; Hübotter et al., 2026; Zhao et al., 2026) The model takes the question and answer as input, and given a one-shot response example, prompt the model to imitate the response's styling. We put our system prompt in Appendix. D.1, and we note this teacher decoding's probability as $p_I(x_T|x_{<T}; Q, A)$.

It is evident that such a baseline can preserve the correctness of the original answers but cannot perfectly replicate the style of the model, as illustrated in Figure 5. While the model's original generation with probability marked as $p_m(x_T|x_{<T}; Q)$ (or student decoding) aligns with its distribution, it does not always yield correct results. This

self-distillation baseline is very straightforward and easy to conceive, so it was explored by various companies during the early models' instruction following fine-tuning phase, before RL was widely applied in LLM post-training. However, it did not see widespread use in the post-training nowadays because its performance was unsatisfactory. Both the case studies in Figure 5 and our quantitative analysis in Section 4.3 show that the responses generated by this approach, even if they appear stylistically similar to the human view, are not truly in-domain from a statistical perspective (or from the LLM's perspective). Our goal is to address this issue.

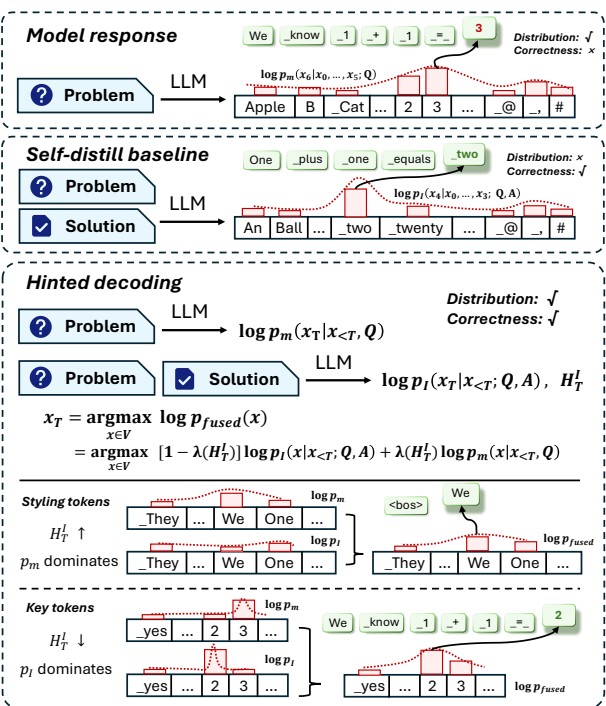

*Figure 6.* The main process of Hinted Decoding.

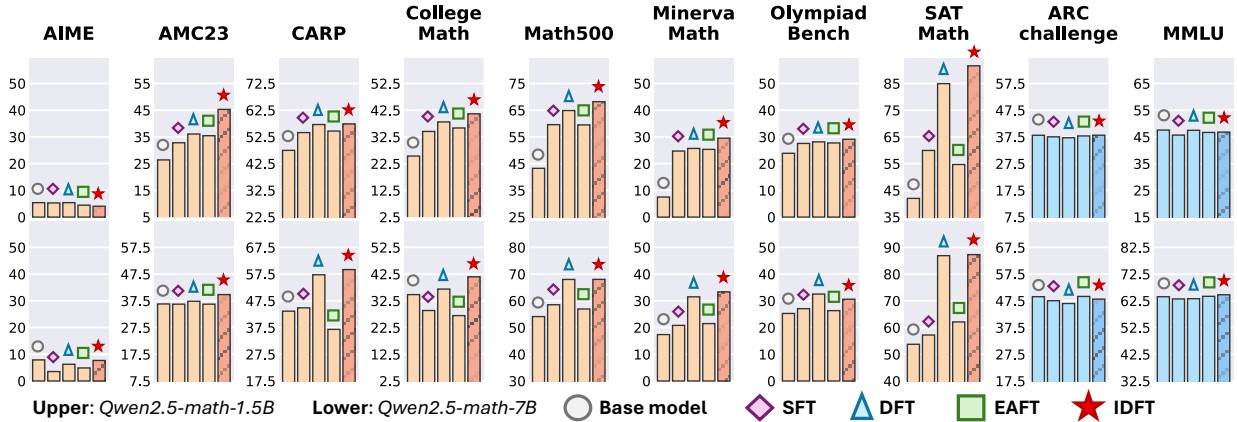

Upper: *Qwen2.5-math-1.5B*   Lower: *Qwen2.5-math-7B*   ⬡ Base model   ◇ SFT   △ DFT   ▢ EAFT   ★ IDFT

*Figure 7.* Comparing IDFT to other enhanced SFT works. Specific evaluation result numbers can be found in Appendix B.3.

**Hinted Decoding.** The central idea of hinted decoding is to dynamically balance between the self-distillation baseline, we call it an imitator: $p_I(x_T|x_{<T}; Q, A)$, and the original model: $p_m(x_T|x_{<T}; Q)$, we hope to maximize $\varphi_m$ (the $\varphi$ of $p_m$), while keeping the answer correct. Specifically, we formulate the following variational problem:

$$\max_q \quad -KL(q||p_I) + E_q[\hat{\lambda}(H^I)\varphi_m], \quad (4)$$

where $q$ is also a distribution function on LLM's vocabulary and $\hat{\lambda}(\cdot)$ is a non-decreasing function. This problem tends to allow the main model to independently handle tokens with high entropy $H^I$, which exhibit high uncertainty and typically determine style, as manifested by the Imitator. In contrast, at critical positions where $H^I$ is low, the Imitator dominate the decoding.

Performing a variational computation on Eq. 4 (details can be found in Appendix A.3), we readily obtain:

$$q \propto p_m^{\hat{\lambda}(H^I)} \cdot p_I.$$

In other words, the target decoding is a weighted combination of the two approaches. For convenience, Hinted Decoding employs the following equivalent logarithmic probabilities for decoding:

$$\log p_{fused} := \left[1 - \lambda(H^I)\right]\log p_I + \lambda(H^I)\log p_m, \quad (5)$$

where we use $\lambda(x) = clip(\beta \cdot x, 0, 1)$ and $\beta$ is the only hyper-parameters that controls the accuracy-distribution trade-off. Figure 6 illustrates the process of Hinted Decoding. We emphasize two pivotal technical details in Hinted Decoding that determine its success or failure:

- **Adaptive mode switching.** While Hinted Decoding can be expressed mathematically as a weighted combination of two modes, its behavior is far from trivial. We experimented with using a constant weighting parameter (setting $\lambda(H^I) = C \in [0, 1]$), but the results were poor (see the ablation in Sec. 4.3). Incorporating $H^I$ into the weighting coefficient is essential: in practice, Hinted Decoding

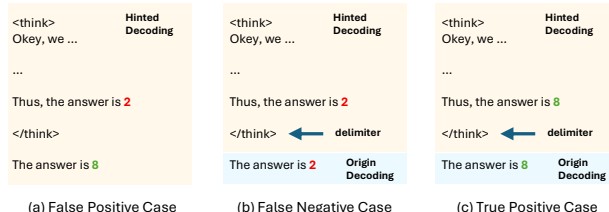

*Figure 8.* (a) Without FP detection mechanism, HD may decode inconsistent response. (b) and (c): Switch back to the student decoding after delimiter ensures the consistency.

functions as a mode switch that increases the teacher's contribution on critical tokens.

- **False-positive (FP) detection mechanism.** We find that applying Hinted Decoding throughout the entire decoding process can lead to inconsistencies between the chain-of-thought (CoT) and the final answer, as shown in the left panel of Fig. 8; we refer to this as the FP case. Although we cannot fully resolve this issue, we can detect such instances and remove them from the dataset. Specifically, Hinted Decoding monitors the delimiter separating the CoT from the final answer (e.g., '$</think>$', '$boxed$'). Once this delimiter is generated, we switch to standard student-only decoding. This preserves CoT–answer consistency; i.e., if the CoT is incorrect, the answer will also be incorrect and can thus be detected and filtered out.

Figure 5 presents the results decoded using Hinted Decoding on real data. It is evident that, despite the simplicity of the approach, it successfully decodes results that maintain correctness while adhering to the model's distribution.

## 4. Experiments

### 4.1. Finetuning loss comparison

We conducted experimental comparisons with other methods aimed at enhancing SFT generalization and performed extensive testing of the results to validate the IDFT loss

*Table 1.* Comparative results of our method, improved SFT, and offline RL. During testing, we used a temperature of 0.3 and we averaged all evaluation results over 16 runs. "***Math-C***" is the average results over three competition benchmarks: AIME24 (Codeforces), AMC23 and Olympiadbench (He et al., 2024). "***Math-G***" is the average of 3 more general math benchmarks: College-math (Tang et al., 2024), Math-OAI (Lightman et al., 2023) and Minerva-math (Dyer & Gur-Ari, 2022). We also test the general reasoning benchmarks MMLU-stem (Hendrycks et al., 2020) and ARC-challenge (Clark et al., 2018) to indicate catastrophic forgetting, with their average number marked as "***General-G***". We also reproduced the evaluation of results in this table, the maximum variance is under $\pm 0.05$. More details can be found in Appendix B. **Bold**: best result, *Italic & underline*: second best results.

| Method | Qwen2.5-7B (base) | | | | Qwen2.5-7B-instruct (instruct) | | | | DeepSeek-R1-distill-Qwen-7B (thinking) | | | |
|---|---|---|---|---|---|---|---|---|---|---|---|---|
| | Budget ↓ | Evaluation Results (%) ↑ | | | Budget ↓ | Evaluation Results (%) ↑ | | | Budget ↓ | Evaluation Results (%) ↑ | | |
| | (GPU Hour) | Math-C | Math-G | General-R | (GPU Hour) | Math-C | Math-G | General-R | (GPU Hour) | Math-C | Math-G | General-R |
| Origin | 0 | 22.06 | 42.13 | 48.28 | 0 | 33.48 | 52.32 | 61.80 | 0 | 43.59 | 54.68 | 59.38 |
| *Supervised finetuning approach (distill DeepSeek-R1)* | | | | | | | | | | | | |
| SFT | 32.85 | 23.93 | 44.24 | 52.10 | 33.25 | 30.07 | 48.33 | 59.93 | 33.04 | 39.43 | 53.52 | 58.28 |
| DFT | 32.84 | 17.09 | 31.34 | 51.93 | 34.01 | 30.22 | 48.16 | 56.08 | 33.49 | 39.52 | 53.03 | 57.39 |
| EAFT | 33.77 | 23.95 | 42.88 | 53.57 | 33.79 | 30.98 | 48.40 | 60.21 | 33.98 | 39.49 | 52.93 | 59.61 |
| *Offline RL approach* | | | | | | | | | | | | |
| Rej@16 | 221.9 | 23.62 | 42.89 | 47.43 | 191.2 | 34.19 | 53.17 | 60.43 | 869.6 | 42.68 | 54.46 | 59.42 |
| DPO@16 | 276.4 | 22.43 | *45.95* | 48.29 | 197.6 | 34.49 | 52.41 | 59.84 | 821.2 | 43.35 | 55.18 | 59.33 |
| SimPO@16 | 230.1 | 20.23 | 38.45 | 47.93 | 190.3 | 34.20 | 52.58 | 60.30 | 801.5 | 43.80 | 54.43 | 59.27 |
| *Baseline and our approach* | | | | | | | | | | | | |
| Self-Distill | 324.7 | 24.96 | 44.07 | 52.99 | 115.7 | 35.09 | 52.39 | 60.25 | 249.3 | 43.74 | *54.90* | 59.79 |
| **HD+SFT** | 212.1 | *25.54* | 45.14 | 52.94 | 139.6 | **36.63** | *53.42* | 60.30 | 424.9 | **44.63** | 54.84 | 60.16 |
| **HD+IDFT** | 214.8 | **27.37** | **47.07** | 53.34 | 135.4 | *36.21* | **53.50** | 60.41 | 425.5 | *43.83* | **55.51** | 59.95 |

proposed in this paper.

**Baselines.** We compared several recent approaches that improve SFT loss to enhance generalization and reduce catastrophic forgetting. All baselines can be represented within the following unified weighting framework generalized as $\mathcal{L} = -w_t \log p_t$.

- ***Standard SFT.*** Standard SFT assigns a static weight $w_t \equiv 1$, imposing uniform supervision across all samples.
- ***Dynamic finetuning (DFT).*** DFT (Wu et al., 2025) sets $w_t = p_t$, which effectively suppresses gradients for OOD data but becomes overly conservative for high-confidence data, resulting in vanishing updates that hinder knowledge consolidation.
- ***Entropy-Adaptive Fine-Tuning (EAFT).*** EAFT (Diao et al., 2026) attempts to gate $w_t$ based on absolute entropy thresholds, it fundamentally conflates context difficulty with distributional alignment, rendering it sensitive to the inherent complexity of the text rather than its correctness.

**Setting.** We trained Qwen2.5-math-1.5B and Qwen2.5-math-7B on Numina-Math (Li et al., 2024) dataset with the baselines mentioned. Unlike previous work, which used identical fixed hyper-parameters for all methods, we fixed all hyper-parameters except for the learning rate. We then conducted a grid search with equal effort to find the optimal learning rate for each method. We believe that using a fixed learning rate for all methods would be unfair, as the improved SFT methods do not normalize the weighting of token loss. For example, consider a naive approach that

weights each token loss by 2.0, this would be approximately equivalent to doubling the learning rate. This naive example demonstrates that algorithms adjusting token loss require different optimal learning rates. Therefore, the most reasonable approach is to search for the optimal learning rate for each method individually. After the training, we evaluate the models with multiple benchmarks, as shown in Figure 7. The evaluation settings are the same as Table 1.

**Results analysis.** The comparative evaluation across ten diverse benchmarks, visualized in Figure 7, highlights the dual superiority of IDFT in facilitating domain-specific acquisition while preserving general capabilities.

In the realm of mathematical reasoning, IDFT consistently outperforms baseline methods (SFT, DFT, and EAFT) across varying difficulty levels. The performance advantage is particularly pronounced on rigorous benchmarks such as Olympiad Bench and AMC23, where logical depth is paramount. This empirical success validates the effectiveness of the Consolidation regime ($\gamma < 1$) within our framework. Unlike DFT, which inadvertently dampens the learning signal for high-confidence tokens due to vanishing gradients, IDFT actively amplifies the updates for these golden reasoning steps. This mechanism ensures that critical logic chains are rigorously mastered rather than merely maintained, enabling the model to solve complex problems where standard baselines falter due to under-fitting. Furthermore, the stability of IDFT over EAFT suggests that the relative distributional statistic $\varphi_t$ offers a more robust filtering criterion than static absolute entropy, effectively distinguishing between necessary mathematical complexity and harmful noise.

Equally significant is the model's performance on general reasoning benchmarks, specifically MMLU and ARC-Challenge. IDFT not only matches but frequently surpasses SFT on these general benchmarks, signaling a successful mitigation of catastrophic forgetting. This robustness is a direct consequence of the Suppression regime ($\gamma > 1$). By automatically identifying and down-weighting OOD tokens ($\varphi \ll 0$), IDFT acts as a selective filter that prevents the model from modifying its parameters to fit noise.

## 4.2. SFT vs Offline RL

In the following contents, we integrate the two methods proposed in this paper to transform a fixed training set into a dataset that aligns with the model's distribution for subsequent training. We then compare most of the well-known offline RL baselines. Additionally, we introduced works that enhance the generalization of SFT as mentioned before.

**Settings.** The experimental setup in this section assumes a fixed-size dataset (DeepMath (He et al., 2025) dataset) to compare the computation-result tradeoff across various methods. Unlike other works that enhance the generalization of SFT by only performing SFT on base models, we employ a stronger baseline by conducting fine-tuning on *Qwen2.5-7B* (base model), *Qwen2.5-7B-instruct* (instruct model) and *DeepSeek-R1-distill-Qwen-7B* (thinking model). This broadens the scope of our experimental validation. Instruct and thinking models have often undergone extensive high-quality SFT and RLHF post-training by their respective organizations, which significantly increases the difficulty of continual learning. Similar to the previous subsection, to ensure a stronger baseline comparison, we conducted a grid search with equal effort to identify the optimal learning rate for each method. Additionally, for methods with specific hyper-parameters, such as DPO-$\beta$, we performed searches for these as well. We kept other less influential hyper-parameters, such as batch size, fixed.

**Baselines.** We include most of the well-known offline RL:

- **Reject sampling finetuning** (Zhang et al., 2023), which is generally considered an implicit RL method without negative samples. During data pre-processing, we let the model roll-out for N times for each sample, and then verify and select the best one into the training set, note as "Rej@N" in Table 1.
- **DPO** (Rafailov et al., 2023) and **SimPO** (Meng et al., 2024). For these preference based RL, we also let the model roll-out N times during data pre-processing. Then we leverage both ground truth verification and LLM to select the preferred and rejected responses.

For SFT methods, since the solutions of DeepMath dataset are verified responses from DeepSeek-R1 (Guo et al., 2025),

the results also represent for distillation approach. For our methods and self-distill baselines, we first let the model generate one rollout for the entire dataset, and for the incorrect samples, we use the corresponding process methods. Then we merge the processed data and the correct rollouts to form the training dataset.

**Results analysis.** We present the results and the computational details of the results in Table 1 and its caption. We summarize the results from the table as follows: (1) When the data distribution deviates from the model's, no improved SFT strategy can completely eliminate the effects of catastrophic forgetting. (2) The comparison between HD+SFT and SFT on general reasoning benchmarks indicates that the data generated to align with the model distribution indeed reduces catastrophic forgetting. (3) Ultimately, our approach consistently surpasses offline RL across each benchmark, while using a lower budget, demonstrating the data efficiency of our approach. (4) Both SFT and self-distillation yield measurable gains on base models. However, as models become more stylistically specialized (e.g., "thinking" models), these baselines deteriorate, whereas our method continues to deliver stable improvements. This further suggests that, although self-distillation may look in-distribution to human observers, it is not necessarily aligned with the model's own data distribution.

Note that these experiments do not indicate that our approach can completely replace RLHF in preference learning and value alignment. This work mainly focuses on scenarios with objective correctness (e.g., mathematics, code, agent).

## 4.3. Ablations

$\beta$ **vs acc** We conducted experiments on the hyperparameter $\beta$ in HD and also validated the performance of HD on large datasets. From a pool of 1000 questions, we used Qwen2.5-7B-instruct to filter out those questions that could not be answered correctly in eight repeated attempts. Fig-

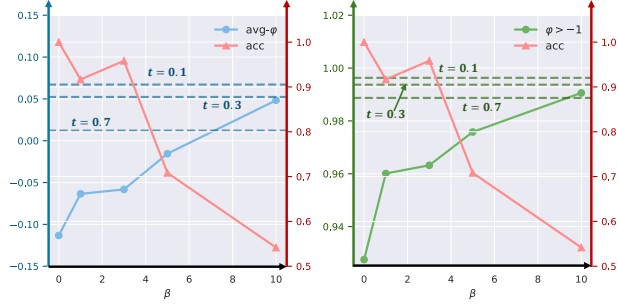

*Figure 9.* The impact of $\beta$.

ure 9 illustrates the variations in accuracy, average $\varphi$, and extreme tokens count under different values of $\beta$ (0, 1, 3, 5, 10) during decoding. Here, $\beta = 0$ serves as the self-distillation baseline. We also plotted the results when the

model solves problems normally, setting the temperature to 0.1, 0.3, and 0.7. The results indicate that when $\beta$ is set to 10, the generated data closely resembles the data produced under normal model conditions.

**Key designs in Hinted Coding**   In Sec. C.1, we introduced two key technical components of Hinted Decoding, the adaptive weighting coefficient and the FP detection mechanism, and we present ablation studies in Table 2. Because in our experiments we first perform a single rollout and apply processing only to problems the model initially answers incorrectly, the appropriate baseline for comparison should be rejection sampling @ 1.

*Table 2.* Ablation for two key components in Hinted Decoding. Numbers in parentheses denote the change relative to the Rej@1 baseline.

| Method | Math-C | Math-G |
|---|---|---|
| Qwen-2.5-7B-instruct | 33.48 | 52.32 |
| + Rej@1 | 33.88 | 52.79 |
| **HD SFT** | **36.63** (**+2.95**) | **53.42** (**+0.63**) |
| $\rightarrow \lambda(H^I) = 0$ (Self-Distill) | 35.09 (+1.21) | 52.39 (−0.40) |
| $\rightarrow \lambda(H^I) = 0.3$ | 34.04 (+0.16) | 52.24 (−0.55) |
| $\rightarrow \lambda(H^I) = 0.5$ | 33.15 (−0.73) | 51.26 (−1.53) |
| $\rightarrow \lambda(H^I) = 0.8$ | 33.78 (−0.10) | 52.03 (−0.76) |
| HD SFT − FP Detection | 35.21 (+1.33) | 52.57 (−0.22) |

The results clearly show that both components have a substantial impact on final performance. A naive weighting scheme is not only highly sensitive to its hyper-parameters but is often detrimental to training. Moreover, without the FP detection mechanism, the method can still deliver some improvement, because the model is reasonably aligned with the data distribution and can tolerate a small amount of noisy data—but it still falls short of the full Hinted Decoding. These findings underscore the importance of the technical details we identified.

**Update component**   To validate the Distribution Discriminant Theory, we conducted an ablation replacing IDFT's soft modulation with a binary gradient mask ($\mathbb{I}[\varphi_t > \tau]$) for $\tau \in \{-1, -5, -10\}$. The results in Table 3 reveal a distinct **inverted-U performance trajectory**.

A strict threshold ($\tau = -1$) degrades performance by excluding informative hard samples ($\varphi_t \in [-5, -1]$), leading to **under-fitting** of complex reasoning patterns. Conversely, a loose threshold ($\tau = -10$) permits high-variance OOD noise ($\varphi_t < -10$), causing **negative transfer** that outweighs data scale benefits. The peak performance at $\tau = -5$ empirically delineates the effective learning interval. IDFT is formulated to align with this topology: its continuous mapping $\gamma_t(\varphi_t)$ adaptively approximates this optimal truncation, preserving gradients for the effective interval ($\varphi \approx 0$) while enforcing attenuation on the distribu-

tion tail ($\varphi \ll 0$) without manual tuning.

*Table 3.* Ablation study on the impact of hard gradient truncation thresholds ($\varphi > \tau$). The **Avg.** column denotes the average accuracy across all six benchmarks. The results demonstrate an inverted-U trend, confirming that an optimal inclusion range exists.

| Method | AIME24 | Math500 | AMC23 | Aqua | College Math | SAT Math | Avg. |
|---|---|---|---|---|---|---|---|
| *Base Model: Qwen Math 1.5B* | | | | | | | |
| Truncate $\varphi > -1$ | 7.93 | 65.60 | 40.31 | 56.18 | **57.84** | **88.48** | 52.72 |
| Truncate $\varphi > -5$ | **8.34** | **67.30** | **41.71** | **67.03** | 57.55 | 87.88 | **54.97** |
| Truncate $\varphi > -10$ | 6.68 | 64.50 | 39.06 | 64.50 | 57.57 | 84.37 | 52.78 |
| *Base Model: Qwen 2.5 7B* | | | | | | | |
| Truncate $\varphi > -1$ | 7.53 | 65.80 | **50.78** | 63.94 | 58.07 | 87.89 | 55.67 |
| Truncate $\varphi > -5$ | **15.20** | **69.70** | 46.25 | **72.50** | 57.32 | **93.79** | **59.13** |
| Truncate $\varphi > -10$ | 10.01 | 69.44 | 40.00 | 48.90 | **59.20** | 87.30 | 52.48 |

## 5. Discussion and Conclusion

Overall, this paper focuses on the distinctions and respective advantages of SFT and RL, aiming to bridge the gap between them by leveraging their strengths. The paper introduces a theoretical framework to quantify and interpret data distribution, and based on this framework, develops two techniques: IDFT and Hinted Decoding, which enhance the generalization capability of SFT.

Due to space constraints, this paper also has its limitations. For example, the algorithm presented here could be adapted into an online version, similar to PPO, where each batch is regenerated, allowing for comparisons with a broader range of RL methods. However, as the research is still in its early stages, we consider this as a future work. Additionally, the algorithm discussed in this paper could demonstrate significantly greater advantages in scenarios, like agents, that those models have not undergone extensive pre-training, compared to its performance in mathematical domains. To ensure the evaluation's credibility, we have also left this aspect for future work.

Last but not least, our work can naturally connect with emerging fields such as speculative decoding (Leviathan et al., 2023), on-policy distillation (Agarwal et al., 2024), and diffusion LLMs (Li et al., 2022; Nie et al., 2025), potentially leading to new algorithms. We believe our work can serve as a baseline and inspire further studies.

## Acknowledgements

This work is supported by the National Natural Science Foundation of China (62576091), the Southeast University Big Data Computing Center, and the Southeast University Kunpeng & Ascend Center of Cultivation.

## Impact Statement

This paper presents work whose goal is to advance the field of Machine Learning. There are many potential societal consequences of our work, none which we feel must be specifically highlighted here.

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

# A. Distribution Discriminant Theory

## A.1. Derivation of Signal-to-Noise Ratio Improvement

In this appendix, we provide a rigorous derivation demonstrating that the proposed Centered Log-Likelihood (CLL) statistic theoretically maximizes the Signal-to-Noise Ratio (SNR) compared to the standard Perplexity (LL) baseline. We employ the *Law of Total Variance* to decompose the uncertainty sources into context-level variability and token-level intrinsic noise.

### A.1.1. PROBLEM SETUP AND DEFINITIONS

Let $c$ denote a context sampled from the data distribution $\mathcal{D}$, and $x$ denote a token generated conditioned on $c$. We define two sources of randomness:

1. **Context Sampling:** $c \sim \mathcal{D}$. The entropy of the model's prediction given $c$ is $H(c) := \mathbb{E}_{x \sim p(\cdot|c)}[-\log p(x|c)]$.

2. **Token Generation:** $x \sim p(\cdot|c)$ under the null hypothesis $\mathcal{H}_0$ (In-Distribution), and $x \sim q(\cdot|c)$ under the alternative hypothesis $\mathcal{H}_1$ (Out-of-Distribution).

We define the *Global Signal-to-Noise Ratio* for a detection statistic $S$ as:

$$\text{SNR}(S) := \frac{(\mathbb{E}_{c,x}[S \mid \mathcal{H}_1] - \mathbb{E}_{c,x}[S \mid \mathcal{H}_0])^2}{\text{Var}_{c,x}(S \mid \mathcal{H}_0)}. \tag{6}$$

### A.1.2. ANALYSIS OF THE BASELINE (LL)

The standard LL statistic is defined as the raw log-likelihood: $S_{\text{LL}}(x, c) = \log p(x|c)$.

### A.1.3. VARIANCE DECOMPOSITION (THE DENOMINATOR)

Using the **Law of Total Variance**, we decompose the variance of $S_{\text{LL}}$ under $\mathcal{H}_0$ into inter-context and intra-context components:

$$\text{Var}(S_{\text{LL}}) = \text{Var}_c\left(\mathbb{E}_{x|c}[S_{\text{LL}}]\right) + \mathbb{E}_c\left[\text{Var}_{x|c}(S_{\text{LL}})\right]. \tag{7}$$

Analyzing the first term (Inter-Context Variance):

$$\mathbb{E}_{x|c}[S_{\text{LL}}] = \mathbb{E}_{x \sim p(\cdot|c)}[\log p(x|c)] = -H(c). \tag{8}$$

Thus, the first term becomes $\text{Var}_c(-H(c)) = \sigma_H^2$, which represents the variability of difficulty across contexts.

Analyzing the second term (Intra-Context Variance): Let $\sigma_\epsilon^2(c) := \text{Var}_{x \sim p(\cdot|c)}(\log p(x|c))$ be the intrinsic aleatoric noise for context $c$. The second term is the average intrinsic noise $\bar{\sigma}_\epsilon^2 = \mathbb{E}_c[\sigma_\epsilon^2(c)]$.

Substituting these back, the total noise for LL is:

$$\text{Noise}_{\text{LL}}^2 = \sigma_H^2 + \bar{\sigma}_\epsilon^2. \tag{9}$$

### A.1.4. SIGNAL ANALYSIS (THE NUMERATOR)

The expected drift under distribution shift is:

$$\begin{aligned}
\Delta_{\text{LL}} &= \mathbb{E}_{c,x}[S_{\text{LL}}|\mathcal{H}_1] - \mathbb{E}_{c,x}[S_{\text{LL}}|\mathcal{H}_0] \\
&= \mathbb{E}_c[-H(q, p)] - \mathbb{E}_c[-H(p)] \\
&= -\mathbb{E}_c[\text{KL}(q(\cdot|c) \,\|\, p(\cdot|c))].
\end{aligned} \tag{10}$$

Let $\Delta = \mathbb{E}_c[\text{KL}(q\|p)]$. The squared signal is $\Delta^2$.

**Resulting SNR for LL:**

$$\text{SNR}_{\text{LL}} = \frac{\Delta^2}{\sigma_H^2 + \bar{\sigma}_\epsilon^2}. \tag{11}$$

A.1.5. ANALYSIS OF PROPOSED METHOD (CLL)

We define the Centered Log-Likelihood as $\varphi(x,c) = \log p(x|c) + H(c)$. Here, $H(c)$ acts as a context-dependent control variate.

A.1.6. VARIANCE REDUCTION

ALLying the Law of Total Variance to $\varphi$ under $\mathcal{H}_0$:

$$\mathrm{Var}(\varphi) = \mathrm{Var}_c\left(\mathbb{E}_{x|c}[\varphi]\right) + \mathbb{E}_c\left[\mathrm{Var}_{x|c}(\varphi)\right]. \tag{12}$$

Crucially, under $\mathcal{H}_0$, the conditional expectation is centered at zero for any context $c$:

$$\mathbb{E}_{x|c}[\log p(x|c) + H(c)] = -H(c) + H(c) = 0. \tag{13}$$

Since the conditional mean is the constant $0$, its variance is zero:

$$\mathrm{Var}_c\left(\mathbb{E}_{x|c}[\varphi]\right) = \mathrm{Var}_c(0) = 0. \tag{14}$$

For the second term, since $H(c)$ is constant given $c$, it does not affect the conditional variance:

$$\mathrm{Var}_{x|c}(\log p(x|c) + H(c)) = \mathrm{Var}_{x|c}(\log p(x|c)) = \sigma_\epsilon^2(c). \tag{15}$$

Thus, the total noise for CLL is significantly reduced:

$$\mathrm{Noise}_{\mathrm{CLL}}^2 = 0 + \bar{\sigma}_\epsilon^2 = \bar{\sigma}_\epsilon^2. \tag{16}$$

A.1.7. SIGNAL PRESERVATION

We verify that the signal magnitude is preserved. The term $H(c)$ cancels out when taking the difference of expectations:

$$\begin{aligned}
\Delta_{\mathrm{CLL}} &= \mathbb{E}[\varphi|\mathcal{H}_1] - \mathbb{E}[\varphi|\mathcal{H}_0] \\
&= (\mathbb{E}[S_{\mathrm{LL}}|\mathcal{H}_1] + \mathbb{E}[H]) - (\mathbb{E}[S_{\mathrm{LL}}|\mathcal{H}_0] + \mathbb{E}[H]) \\
&= \Delta_{\mathrm{LL}} = -\Delta.
\end{aligned} \tag{17}$$

The squared signal remains $\Delta^2$.

A.1.8. CONCLUSION

Comparing the derived SNRs:

$$\mathrm{SNR}_{\mathrm{CLL}} = \frac{\Delta^2}{\bar{\sigma}_\epsilon^2} \quad \text{vs.} \quad \mathrm{SNR}_{\mathrm{LL}} = \frac{\Delta^2}{\sigma_H^2 + \bar{\sigma}_\epsilon^2}. \tag{18}$$

Since natural language datasets exhibit diverse complexity, implying $\sigma_H^2 > 0$, we conclude that:

$$\mathrm{SNR}_{\mathrm{CLL}} > \mathrm{SNR}_{\mathrm{LL}}. \tag{19}$$

This proves that the CLL statistic strictly dominates the LL baseline in terms of Signal-to-Noise Ratio by eliminating the variance contribution from context difficulty.

## A.2. Mathematical Proofs of Sequential Properties

In this section, we provide rigorous derivations for the stochastic properties of the cumulative statistic $S_n = \sum_{t=1}^n \varphi_t$, where $\varphi_t = \log p_t(x_t) + H(p_t)$.

A.2.1. PROOF OF MARTINGALE PROPERTY UNDER $\mathcal{H}_0$

**Proposition A.1.** *Under the in-distribution hypothesis $\mathcal{H}_0$ (where $x_t \sim p_t$), the sequence $\{S_n\}_{n \geq 0}$ is a discrete-time martingale with respect to the filtration $\mathcal{F}_n = \sigma(x_1, \ldots, x_n)$.*

*Proof.* A sequence is a martingale if $\mathbb{E}[|S_n|] < \infty$ and $\mathbb{E}[S_n \mid \mathcal{F}_{n-1}] = S_{n-1}$. Consider the increment $\Delta S_n = S_n - S_{n-1} = \varphi_t$. We calculate its conditional expectation under $\mathcal{H}_0$:

$$
\begin{aligned}
\mathbb{E}[\varphi_t \mid \mathcal{F}_{n-1}, \mathcal{H}_0] &= \mathbb{E}_{x_t \sim p_t}[\log p_t(x_t) + H(p_t)] \\
&= \mathbb{E}_{x_t \sim p_t}[\log p_t(x_t)] + H(p_t) \\
&= \sum_{x \in \mathcal{V}} p_t(x) \log p_t(x) + H(p_t) \\
&= -H(p_t) + H(p_t) \\
&= 0.
\end{aligned}
\tag{20}
$$

Since the expected increment is zero, we have:

$$
\mathbb{E}[S_n \mid \mathcal{F}_{n-1}] = S_{n-1} + \mathbb{E}[\varphi_t \mid \mathcal{F}_{n-1}] = S_{n-1}.
\tag{21}
$$

Thus, $S_n$ is a zero-mean martingale under $\mathcal{H}_0$. $\qquad\square$

### A.2.2. PROOF OF NEGATIVE DRIFT UNDER $\mathcal{H}_1$

**Proposition A.2.** *Under the out-of-distribution hypothesis $\mathcal{H}_1$ (where $x_t \sim q_t \neq p_t$), the statistic $\varphi_t$ has a strictly negative expectation if the KL divergence dominates the entropy gap.*

*Proof.* We calculate the expectation of $\varphi_t$ with respect to the true generating distribution $q_t$:

$$
\begin{aligned}
\mathbb{E}[\varphi_t \mid \mathcal{H}_1] &= \mathbb{E}_{x_t \sim q_t}[\log p_t(x_t) + H(p_t)] \\
&= \sum_{x \in \mathcal{V}} q_t(x) \log p_t(x) + H(p_t).
\end{aligned}
\tag{22}
$$

We rewrite the cross-entropy term $\sum q \log p$:

$$
\begin{aligned}
\sum_x q_t(x) \log p_t(x) &= \sum_x q_t(x) \log \frac{p_t(x)}{q_t(x)} q_t(x) \\
&= \sum_x q_t(x) \log \frac{p_t(x)}{q_t(x)} + \sum_x q_t(x) \log q_t(x) \\
&= -D_{\mathrm{KL}}(q_t \| p_t) - H(q_t).
\end{aligned}
\tag{23}
$$

Substituting this back, we obtain:

$$
\mathbb{E}[\varphi_t \mid \mathcal{H}_1] = -D_{\mathrm{KL}}(q_t \| p_t) + (H(p_t) - H(q_t)).
\tag{24}
$$

Since the KL divergence $D_{\mathrm{KL}}(q_t \| p_t) \geq 0$ is typically the dominant term for OOD samples (where the model assigns low probability to the observed distribution), and assuming the entropy difference is bounded or negligible compared to the divergence, the expectation is negative. Specifically, if $D_{\mathrm{KL}}(q_t \| p_t) > H(p_t) - H(q_t)$, then $\mathbb{E}[\Delta S_n] < 0$, causing a downward linear drift. $\qquad\square$

### A.2.3. CONCENTRATION INEQUALITY AND ERROR BOUNDS

To theoretically bound the False Positive Rate (Type I Error), we analyze the probability that the cumulative score $S_L$ drops below a threshold $-\lambda$ given that the sequence is actually In-Distribution ($\mathcal{H}_0$).

**Theorem A.3** (Tail Bound for IDFT)**.** *Assume the statistic $\varphi_t$ is bounded such that $|\varphi_t| \leq c$ for some constant $c$, and let $\sigma_t^2 = \mathrm{Var}(\varphi_t \mid \mathcal{H}_0)$. For a sequence of length $L$, the probability of a false rejection is bounded by:*

$$
\mathbb{P}(S_L \leq -\lambda \mid \mathcal{H}_0) \leq \exp\left(-\frac{\lambda^2}{2\sum_{t=1}^{L} \sigma_t^2 + \frac{2}{3}c\lambda}\right).
\tag{25}
$$

*Proof.* Since $\{S_n\}$ is a zero-mean martingale with bounded increments $|\varphi_t| \leq c$ and conditional variance $\sigma_t^2$, we can invoke the **Freedman's Inequality** (a martingale variant of Bernstein's inequality). Freedman's inequality states that for a martingale difference sequence $X_t$ with $X_t \leq c$, and $V_L = \sum \mathrm{Var}(X_t \mid \mathcal{F}_{t-1})$, the tail probability is bounded by:

$$\mathbb{P}\left(\sum_{t=1}^{L} X_t \geq \lambda\right) \leq \exp\left(-\frac{\lambda^2}{2V_L + 2c\lambda/3}\right). \tag{26}$$

ALLying this to our statistic: we are interested in the lower tail $S_L \leq -\lambda$. By symmetry of the bound (or aLLying to $-S_L$), and noting that $V_L = \sum_{t=1}^{L} \sigma_t^2$, we directly obtain the stated bound.

This result implies that the false positive rate decays exponentially with the squared threshold $\lambda^2$, scaled by the accumulated aleatoric variance $\sum \sigma_t^2$. Compared to standard raw log-likelihood (which includes the high variance of entropy $\sigma_H^2$ in the denominator), our statistic $\varphi_t$ minimizes the denominator to $\sum \sigma_\epsilon^2$, thereby significantly tightening the error bound and allowing for more sensitive detection thresholds. $\square$

### A.3. Derivation of the Formula for Hinted Decoding

We start from the variational problem:

$$\max_q \quad -KL(q||p_I) + E_q[\hat{\lambda}(H^I)\varphi_m],$$

Using the definition:

$$\begin{aligned}
\mathcal{J}[q] &= -KL(q||p_I) + E_q[\hat{\lambda}(H^I)\varphi_m], \\
&= -\int q \cdot \log \frac{q}{p_I} + \int q \cdot \hat{\lambda}(H^I)\left(\log p_m + H^m\right) \\
&= \int q \cdot \left[\log p_I + \hat{\lambda}(H^I)\log p_m - \log q\right] + \hat{\lambda}(H^I)H^m
\end{aligned}$$

The objective functional $\mathcal{J}[q]$ is a strictly concave function with respect to $q$ (when $q$ is at an interior point), because $\int q \log q$ is strictly concave. The linear term does not alter the concavity, therefore, the global optimum is uniquely determined by the first-order condition. The Lagrangian function with the normalization constraint can be expressed as follows. Given the constraint $\int q = 1$, the Lagrangian function is defined as:

$$\mathcal{L}[q, \alpha] = \mathcal{J}[q] + \alpha \cdot \left(\int q - 1\right).$$

Now, we take the partial derivative of the Lagrangian function with respect to each component and set it to zero:

$$\frac{\partial \mathcal{L}}{\partial q} = \hat{\lambda}(H^I)\log p_m + \log p_I - \log q - 1 + \alpha = 0,$$

which gives:

$$q = C \cdot p_I \cdot p_m^{\hat{\lambda}(H^I)}.$$

### A.4. Theoretical Justification for the Logarithmic Transformation

In this appendix, we provide a comprehensive justification for selecting the logarithmic function as the basis for our detection statistic. We first present the physical intuition grounded in the mechanics of neural networks, and then provide a rigorous algebraic proof based on group isomorphism.

A.4.1. PHYSICAL MOTIVATION: MULTIPLICATIVE NOISE IN LLMS

The assumption that aleatoric uncertainty in language models is multiplicative (rather than additive) is grounded in the fundamental architectural constraints of modern neural networks.

**The Softmax Amplification Mechanism.** Neural networks typically operate in a continuous logit space $z \in \mathbb{R}^V$, where internal perturbations (arising from quantization errors, dropout, or layer norm fluctuations) act additively:

$$z_{obs} = z_{true} + \epsilon, \tag{27}$$

where $\epsilon$ represents stochastic noise in the latent representation. However, the final probability distribution is generated via the Softmax function $p_i \propto e^{z_i}$. Consequently, an additive perturbation in the logit space manifests as a multiplicative factor in the probability space:

$$p_{obs} \propto e^{z_{obs}} = e^{z_{true}+\epsilon} = e^{z_{true}} \cdot e^{\epsilon} = p_{true} \cdot \xi, \tag{28}$$

where $\xi = e^{\epsilon}$. This structural property inherent to LLMs dictates that the magnitude of noise scales proportionally with the probability magnitude itself, establishing the multiplicative nature of the uncertainty.

**Scale Constraints of the Probability Simplex.** Assuming an additive noise model (e.g., $p_{obs} = p_{true} + \delta$, where $\delta$ is fixed) leads to physical inconsistencies, particularly for the long-tail distribution of tokens. Consider a rare token with a true probability $p_{true} \approx 10^{-9}$. A standard fixed additive noise (e.g., $\delta = 10^{-4}$) would imply that:

- The observed probability could shift to negative values ($p < 0$), violating the axioms of probability.

- A negligible noise term could arbitrarily increase the probability of a rare token by orders of magnitude (e.g., from $10^{-9}$ to $10^{-4}$), effectively destroying the semantic structure of the language model.

In contrast, a multiplicative noise model ($p_{obs} = p_{true} \cdot \xi$) inherently respects the boundary conditions of the probability simplex. It ensures that rare events remain rare under stochastic fluctuation, preserving the order of magnitude for both high-frequency and low-frequency tokens.

A.4.2. EXACT DERIVATION VIA GROUP ISOMORPHISM

Based on the multiplicative noise model established above, we seek a transformation that stabilizes the variance across the entire vocabulary, enabling a unified detection threshold.

**Problem Formulation.** Let the observed probability be modeled as $p_{obs} = p_{true} \cdot \xi$, where $\xi$ is a random noise variable independent of the signal strength $p_{true}$. In the linear probability space, the variance is heteroscedastic:

$$\mathrm{Var}(p_{obs} \mid p_{true}) = \mathrm{Var}(p_{true} \cdot \xi) = p_{true}^2 \cdot \mathrm{Var}(\xi). \tag{29}$$

This quadratic scaling ($\propto p^2$) creates a "scale collapse" effect, where the variance for high-probability tokens dominates the detection metric, rendering it insensitive to anomalies in the low-probability regime.

**Variance Stabilization via Isomorphism.** We rely on the *translation-invariance property* of variance: $\mathrm{Var}(Y + c) = \mathrm{Var}(Y)$, where $c$ is a constant. To utilize this property for stabilization, we require a transformation $f(\cdot)$ that maps the multiplicative structure of the noise to an additive structure. Mathematically, this requires a group isomorphism from the multiplicative group of positive real numbers $(\mathbb{R}^+, \times)$ to the additive group of real numbers $(\mathbb{R}, +)$:

$$f(x \cdot y) = f(x) + f(y). \tag{30}$$

The logarithmic function $f(p) = \log p$ is the unique continuous solution to this functional equation (up to a scaling constant).

**Proof of Homoscedasticity.** Applying this transformation to our noise model:

$$\log p_{obs} = \log(p_{true} \cdot \xi) = \log p_{true} + \log \xi. \tag{31}$$

We now compute the variance of the transformed statistic conditioned on the context (where $p_{true}$ is fixed):

$$\begin{aligned}
\mathrm{Var}(\log p_{obs} \mid p_{true}) &= \mathrm{Var}(\log p_{true} + \log \xi) \\
&= \mathrm{Var}(\log \xi).
\end{aligned} \tag{32}$$

Crucially, the term $\log p_{true}$ acts as an additive constant and vanishes during the variance calculation. The resulting variance $\text{Var}(\log \xi)$ depends *solely* on the intrinsic noise distribution $\xi$ and is invariant to the probability magnitude $p_{true}$.

*Conclusion:* This exact derivation demonstrates that $\log p$ is the theoretically optimal transformation for normalizing multiplicative noise. It ensures that the noise floor (denominator in SNR) is uniform across the entire vocabulary, thereby strictly justifying the choice of the logarithmic scale in our proposed statistic.

## A.5. Empirical Validation

### A.5.1. QWEN

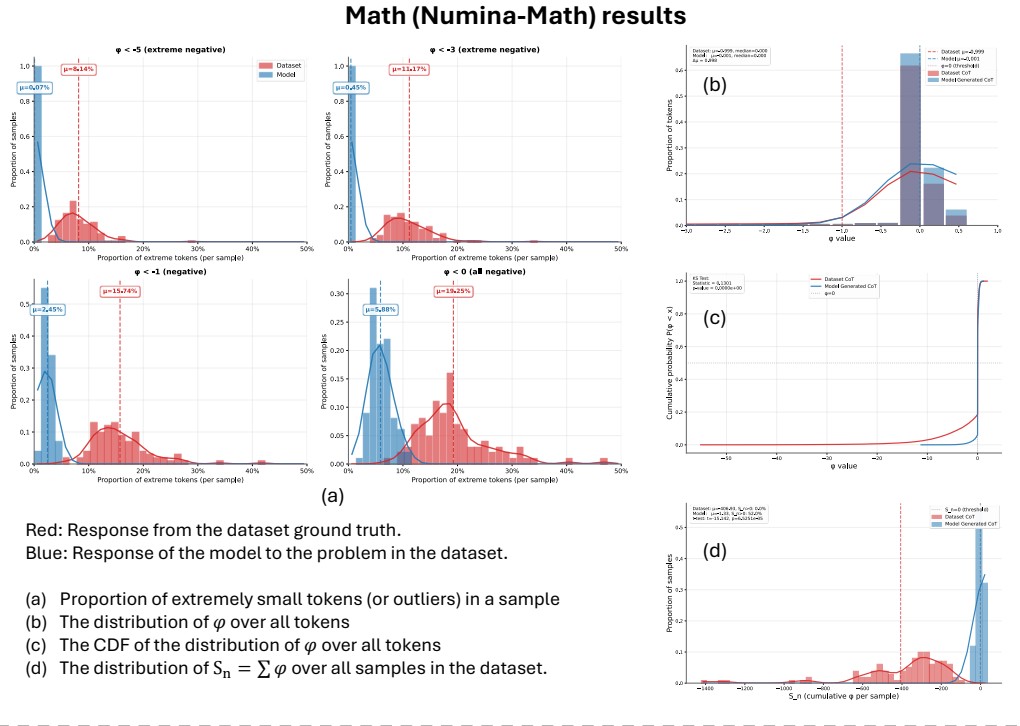

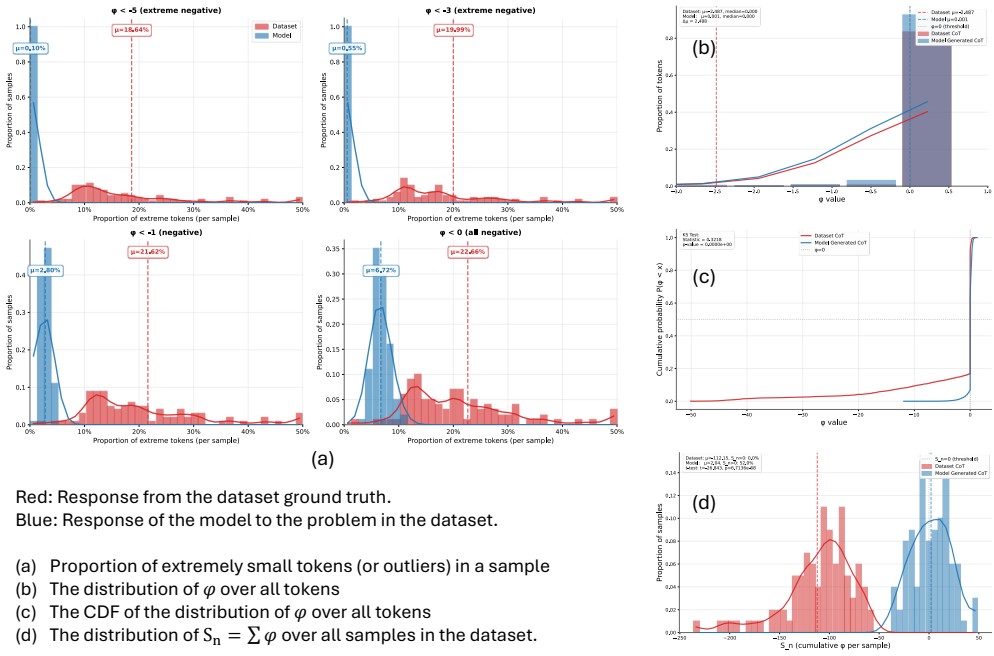

*Figure 10.* The statistic results with **Qwen3-8B** model.

A.5.2. DEEPSEEK

## Math (Numina-Math) results

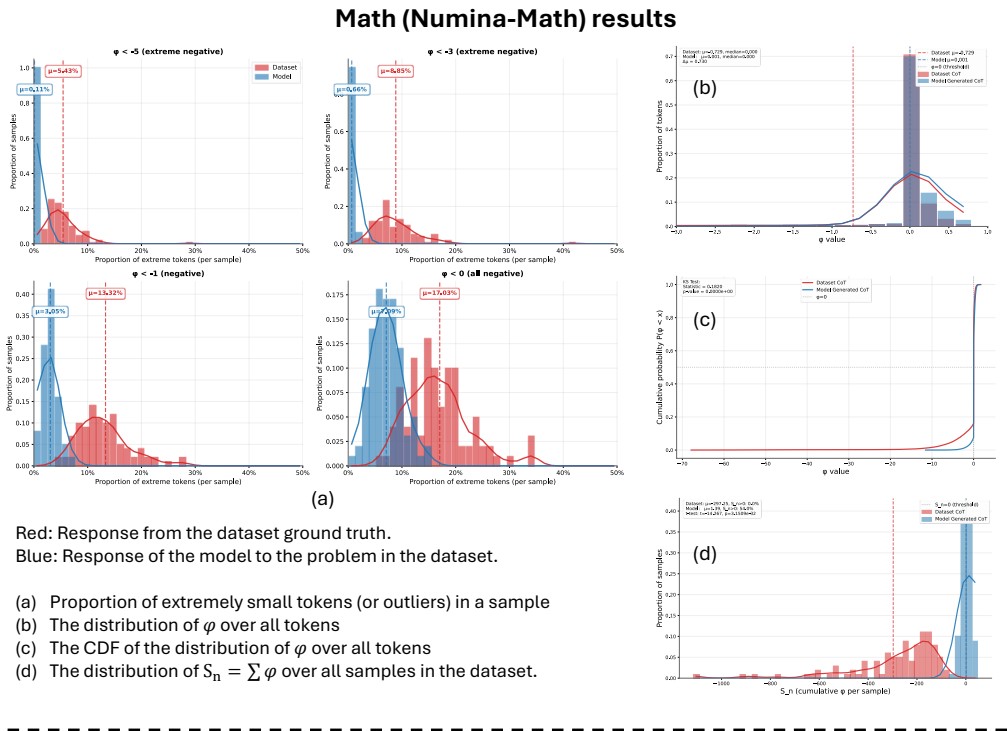

Red: Response from the dataset ground truth.
Blue: Response of the model to the problem in the dataset.

(a) Proportion of extremely small tokens (or outliers) in a sample
(b) The distribution of $\varphi$ over all tokens
(c) The CDF of the distribution of $\varphi$ over all tokens
(d) The distribution of $S_n = \sum \varphi$ over all samples in the dataset.

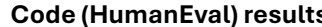

## Code (HumanEval) results

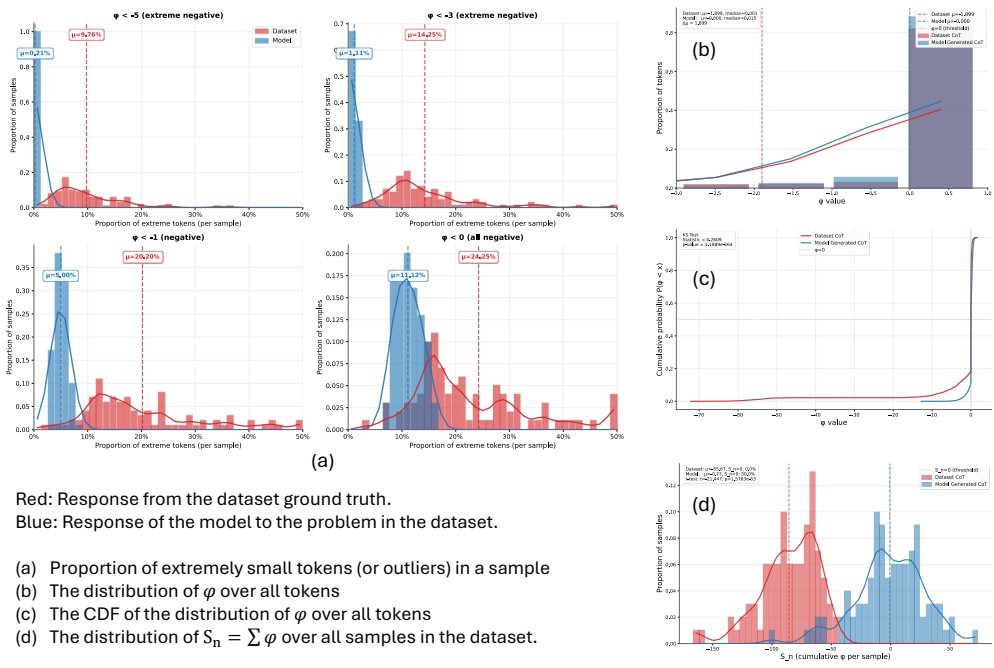

Red: Response from the dataset ground truth.
Blue: Response of the model to the problem in the dataset.

(a) Proportion of extremely small tokens (or outliers) in a sample
(b) The distribution of $\varphi$ over all tokens
(c) The CDF of the distribution of $\varphi$ over all tokens
(d) The distribution of $S_n = \sum \varphi$ over all samples in the dataset.

*Figure 11.* The statistic results with **Deepseek-R1** model.

## A.5.3. MISTRAL

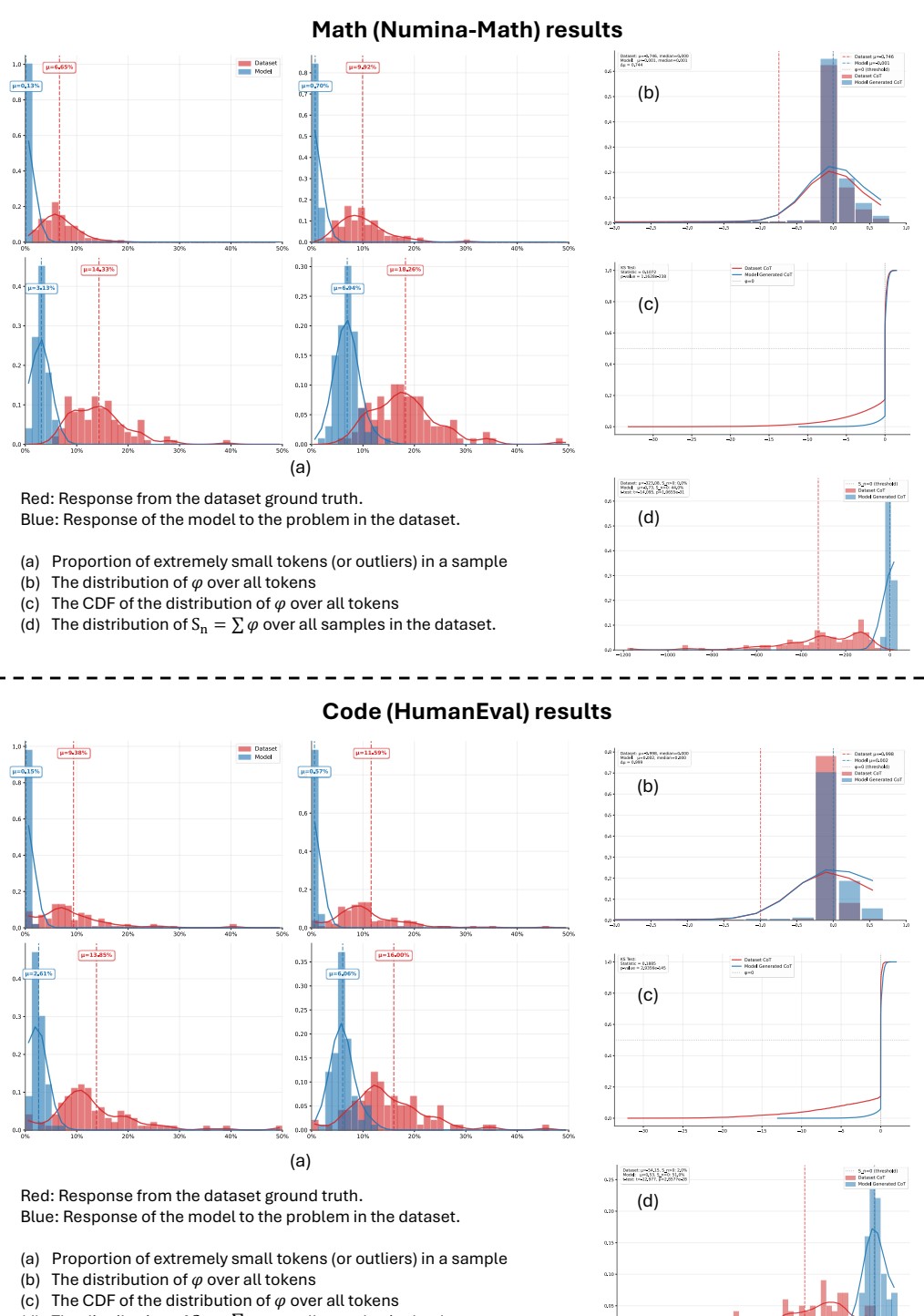

*Figure 12.* The statistic results with **Mistral-7B** model.

## A.5.4. LLAMA

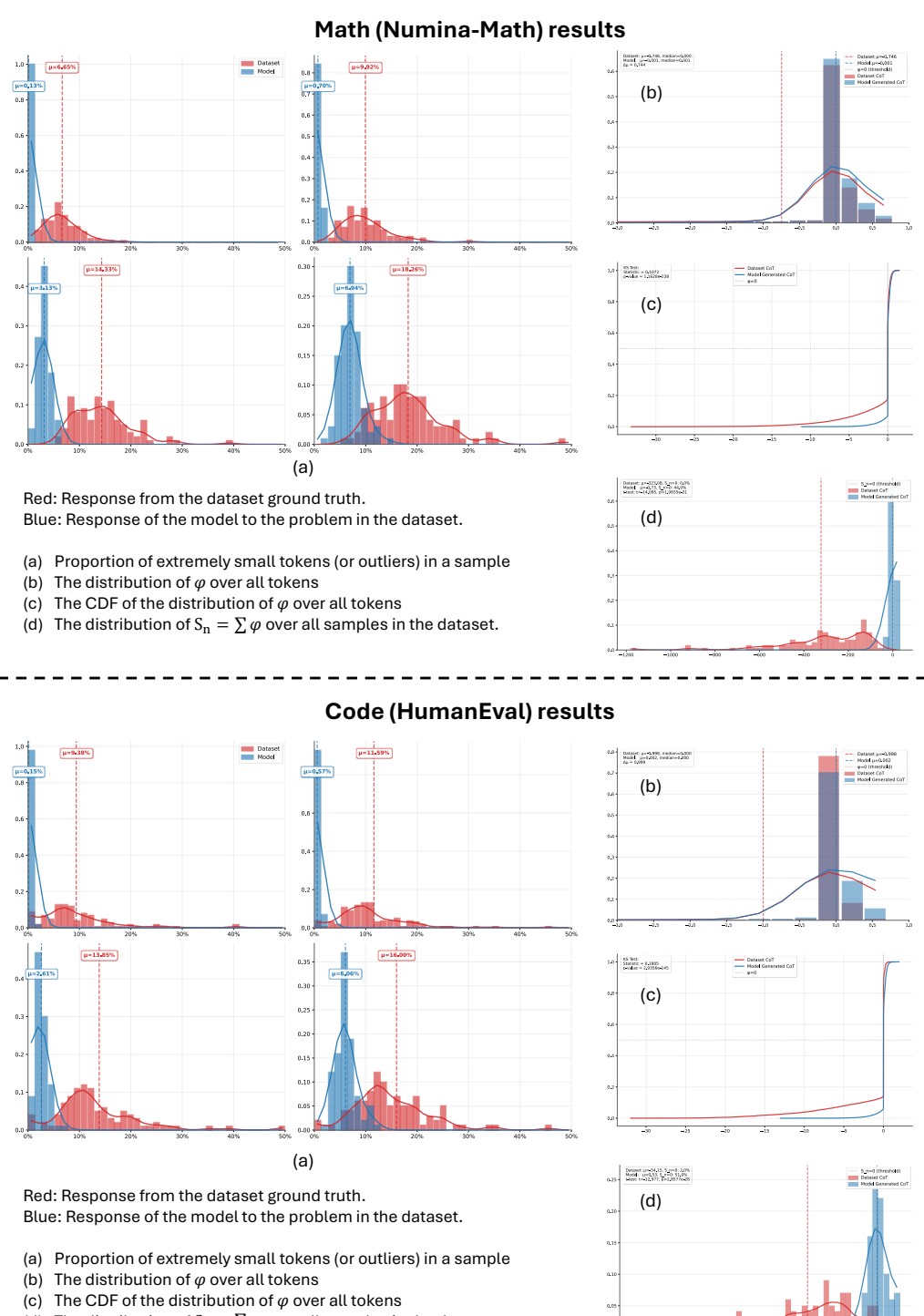

### Math (Numina-Math) results

Red: Response from the dataset ground truth.
Blue: Response of the model to the problem in the dataset.

(a) Proportion of extremely small tokens (or outliers) in a sample
(b) The distribution of $\varphi$ over all tokens
(c) The CDF of the distribution of $\varphi$ over all tokens
(d) The distribution of $S_n = \sum \varphi$ over all samples in the dataset.

### Code (HumanEval) results

Red: Response from the dataset ground truth.
Blue: Response of the model to the problem in the dataset.

(a) Proportion of extremely small tokens (or outliers) in a sample
(b) The distribution of $\varphi$ over all tokens
(c) The CDF of the distribution of $\varphi$ over all tokens
(d) The distribution of $S_n = \sum \varphi$ over all samples in the dataset.

*Figure 13.* The statistic results with **LLaMA-3.1-8B** model.

# B. More Experiment Details

## B.1. Benchmarks selection

The benchmarks listed in Table 1 were not cherry-picked to favor our approach, instead, we evaluated all the following benchmarks: AMC23, AIME24, AIME25, CARP, College-math, MATH, Math-OAI, Minerva-math, Olympiadbench, SAT-math, AQUA, GSM8K. We observed that for several of these benchmarks, even using Offline RL algorithms failed to yield performance improvements. This is primarily due to the use of a strong baseline model. Instruct models are typically fine-tuned by various organizations on their respective high-quality datasets, which often include extensive mathematical datasets, and many of these models have also undergone RL post-training via RLHF. Consequently, the degree of overlap between the training and test datasets becomes a key factor influencing performance. Therefore, after evaluating the aforementioned benchmarks, we selected all benchmarks that satisfied the following criterion: at least two Offline RL algorithms were able to stably achieve performance improvements on the benchmark. The selected benchmarks are those presented in Table 1.

## B.2. Budget Calculation

We implemented Hinted Decoding in vLLM and used the vLLM library for all generations. We report aggregate GPU-hours, including the time spent by an equally sized LLM used to assess answer correctness. Notably, for rollouts such as DPO@16, we did not inflate runtime by sampling 16 candidates per instance in one shot. Instead, we adopted the most compute-efficient protocol: generate one sample at a time and roll out an additional sample only when the current one fails to meet the criterion (up to the specified cap). All reported runtimes were measured on NVIDIA H100 machines equipped with NVLink. We do not report FLOPs-based metrics because generation is memory-bound rather than compute-bound; FLOPs would substantially understate the actual wall-clock generation time.

## B.3. Detailed Evaluation Results

*Table 4.* Ablation-The impact of $\beta$

| Temperature | 0.1 | 0.3 | 0.7 | $\beta =$ | 0 | 1 | 3 | 5 | 10 |
|---|---|---|---|---|---|---|---|---|---|
| Accuracy (%) ↑ | 0.0 | 0.0 | 0.0 | | 100.0 | 91.7 | 95.8 | 70.8 | 54.2 |
| AVG-$\psi$ ↑ | 0.0671 | 0.0523 | 0.0121 | | -0.1132 | -0.0635 | -0.0582 | -0.0155 | 0.0483 |
| $\psi > -1$ (%) ↑ | 99.63 | 99.37 | 98.87 | | 92.75 | 96.01 | 96.32 | 97.58 | 99.06 |
| $\psi > -3$ (%) ↑ | 100.00 | 99.90 | 99.17 | | 96.69 | 97.18 | 97.48 | 97.90 | 99.65 |
| $\psi > -5$ (%) ↑ | 100.00 | 99.97 | 99.42 | | 97.74 | 99.36 | 99.35 | 99.57 | 99.85 |

*Table 5.* Detailed evaluation results of Qwen2.5-7B-Base

| Method | Qwen2.5-7B-Base | | | | | | | | | | | |
|---|---|---|---|---|---|---|---|---|---|---|---|---|
| | AMC23 | AIME24 | Oliympiadbench | Math-C | College-Math | Math-OAI | Minerva-math | Math-G | MMLU-Stem | ARC-Challenge | General-R | AVG |
| Qwen base | 29.84 | 6.86 | 29.50 | 22.07 | 42.03 | 60.50 | 23.88 | 42.14 | 56.89 | 39.67 | 48.28 | 36.51 |
| | *Supervised finetuning approach* | | | | | | | | | | | |
| Origin SFT | 35.31 | 8.75 | 27.73 | 23.93 | 40.09 | 65.02 | 27.60 | 44.24 | 66.21 | 37.99 | 52.10 | 39.00 |
| Origin DFT | 30.15 | 3.32 | 17.80 | 17.09 | 30.03 | 44.83 | 19.16 | 31.34 | 65.91 | 37.95 | 51.93 | 31.77 |
| Origin EAFT | 36.25 | 8.95 | 26.67 | 23.96 | 38.87 | 63.15 | 26.63 | 42.88 | 69.45 | 37.69 | 53.57 | 38.92 |
| | *Offline RL approach* | | | | | | | | | | | |
| Reject sampling | 35.15 | 7.93 | 27.77 | 23.62 | 42.95 | 62.57 | 23.14 | 42.89 | 56.29 | 38.56 | 47.43 | 37.12 |
| DPO | 32.50 | 5.84 | 28.94 | 22.43 | 44.45 | 63.60 | 29.79 | 45.95 | 56.83 | 39.76 | 48.30 | 38.03 |
| SimPO | 29.84 | 5.83 | 25.04 | 20.24 | 37.02 | 53.17 | 25.16 | 38.45 | 56.65 | 39.21 | 47.93 | 34.41 |
| | *Our approach* | | | | | | | | | | | |
| SD+SFT | 36.09 | 8.54 | 30.26 | 24.96 | 43.17 | 64.37 | 24.68 | 44.07 | 66.48 | 39.50 | 52.99 | 39.56 |
| HD+SFT | 38.75 | 7.70 | 30.16 | 25.54 | 44.35 | 65.30 | 25.78 | 45.14 | 66.54 | 39.33 | 52.94 | 40.14 |
| HD+IDFT | 41.88 | 8.75 | 31.48 | 27.37 | 43.33 | 66.40 | 31.48 | 47.07 | 66.83 | 39.84 | 53.34 | 41.61 |

*Table 6.* Detailed evaluation results of Qwen2.5-7B-instruct model

| Method | AMC23 | AIME24 | Oliympiadbench | Math-C | College-Math | Math-OAI | Minerva-math | Math-G | MMLU-Stem | ARC-Challenge | General-R | AVG |
|---|---|---|---|---|---|---|---|---|---|---|---|---|
| | **Qwen2.5-7B-instruct** | | | | | | | | | | | |
| origin | 51.7 | 10.6 | 38.5 | 33.6 | 43.4 | 76.0 | 41.0 | 53.5 | 68.3 | 55.2 | 61.8 | 49.6 |
| | *Supervised finetuning approach* | | | | | | | | | | | |
| SFT | 24.1 | 2.3 | 19.0 | 15.1 | 24.2 | 41.8 | 10.0 | 25.3 | 62.7 | 51.0 | 56.9 | 32.4 |
| DFT | 16.1 | 2.3 | 13.8 | 10.7 | 24.7 | 34.4 | 9.5 | 22.8 | 56.0 | 45.0 | 50.5 | 28.0 |
| EAFT | 30.0 | 3.1 | 17.5 | 16.9 | 25.0 | 44.7 | 11.3 | 27.0 | 63.3 | 48.6 | 55.9 | 33.3 |
| | *Offline RL approach* | | | | | | | | | | | |
| Rej@2 | 50.8 | 10.1 | 37.9 | 32.9 | 43.4 | 75.7 | 41.1 | 53.4 | 68.4 | 54.0 | 61.2 | 49.2 |
| Rej@4 | 49.2 | 10.2 | 38.8 | 32.7 | 43.8 | 76.0 | 42.5 | 54.1 | 68.9 | 53.8 | 61.4 | 49.4 |
| DPO | 51.4 | 11.0 | 38.3 | 33.6 | 43.2 | 75.7 | 41.7 | 53.5 | 68.2 | 55.0 | 61.6 | 49.6 |
| RPO | 50.8 | 10.6 | 35.9 | 32.4 | 42.2 | 73.8 | 40.4 | 52.1 | 68.5 | 55.1 | 61.8 | 48.8 |
| SimPO | 53.4 | 10.6 | 37.5 | 33.9 | 43.5 | 77.0 | 41.8 | 54.1 | 68.5 | 55.1 | 61.8 | 49.9 |
| CPO | 51.9 | 11.5 | 37.2 | 33.5 | 43.4 | 76.4 | 41.4 | 53.7 | 68.5 | 54.3 | 61.4 | 49.5 |
| | *Our approach* | | | | | | | | | | | |
| HD+SFT | 53.8 | 10.2 | 38.2 | 34.1 | 43.6 | 75.5 | 42.5 | 53.9 | 68.5 | 53.8 | 61.2 | 49.7 |
| HD+IDFT | 53.4 | 12.3 | 38.0 | 34.6 | 42.0 | 76.2 | 41.3 | 53.2 | 68.4 | 54.6 | 61.5 | 49.8 |

*Table 7.* Detailed evaluation results of DeepSeek-R1-Distill-Qwen-7B

| Method | AMC23 | AIME24 | Oliympiadbench | Math-C | College-Math | Math-OAI | Minerva-math | Math-G | MMLU-Stem | ARC-Challenge | General-R | AVG |
|---|---|---|---|---|---|---|---|---|---|---|---|---|
| | **DeepSeek-R1-Distill-Qwen-7B** | | | | | | | | | | | |
| origin | 63.90 | 25.61 | 41.28 | 43.60 | 44.77 | 83.30 | 35.97 | 54.68 | 70.56 | 48.20 | 59.38 | 51.93 |
| | *Supervised finetuning approach* | | | | | | | | | | | |
| SFT | 60.63 | 18.75 | 38.90 | 39.43 | 44.73 | 80.58 | 35.25 | 53.52 | 67.64 | 52.21 | 59.93 | 50.14 |
| DFT | 59.36 | 19.21 | 40.01 | 39.53 | 44.59 | 80.40 | 34.12 | 53.04 | 66.00 | 46.16 | 56.08 | 48.95 |
| EAFT | 60.44 | 18.88 | 39.17 | 39.50 | 44.20 | 79.44 | 35.17 | 52.94 | 67.26 | 53.15 | 60.21 | 50.03 |
| | *Offline RL approach* | | | | | | | | | | | |
| Reject sampling | 63.75 | 23.11 | 41.19 | 42.68 | 44.40 | 83.03 | 35.95 | 54.46 | 70.47 | 48.37 | 59.42 | 51.53 |
| DPO | 63.59 | 25.00 | 41.46 | 43.35 | 44.90 | 84.07 | 36.57 | 55.18 | 70.37 | 48.29 | 59.33 | 52.01 |
| SimPO | 64.21 | 25.00 | 42.19 | 43.80 | 44.59 | 83.07 | 35.63 | 54.43 | 70.34 | 48.20 | 59.27 | 51.88 |
| | *Our approach* | | | | | | | | | | | |
| SD+SFT | 64.54 | 24.78 | 41.90 | 43.74 | 45.10 | 82.95 | 36.66 | 54.90 | 68.37 | 51.22 | 59.80 | 52.18 |
| HD+SFT | 65.31 | 27.09 | 41.48 | 44.63 | 45.21 | 83.15 | 36.16 | 54.84 | 68.53 | 51.79 | 60.16 | 52.58 |
| HD+IDFT | 64.22 | 26.25 | 41.03 | 43.83 | 45.38 | 84.15 | 36.99 | 55.51 | 68.44 | 51.45 | 59.95 | 52.47 |

*Table 8.* Performance comparison on Qwen2.5-Math-1.5B and Qwen2.5-Math-7B.

| Method | **Math Reasoning** | | | | | | | | **General** | | AVG |
|---|---|---|---|---|---|---|---|---|---|---|---|
| | AIME | AMC23 | CARP | College | MATH500 | Minerva | Olympiad | SAT | ARC | MMLU | |
| | *Qwen2.5-Math-1.5B* | | | | | | | | | | |
| Base Model | 5.41 | 26.40 | 47.47 | 25.35 | 43.25 | 7.50 | 23.88 | 42.00 | 38.13 | 47.57 | 30.7 |
| SFT | 5.31 | 32.81 | 54.13 | 34.55 | 59.60 | 24.71 | 27.57 | 59.95 | 37.54 | 45.68 | 38.2 |
| DFT | 5.41 | 36.09 | 57.14 | 38.11 | 64.88 | 25.70 | 28.15 | 84.95 | 37.20 | 47.48 | 42.5 |
| EAFT | 4.47 | 35.46 | 54.70 | 35.85 | 59.50 | 25.34 | 27.75 | 54.69 | 37.88 | 46.68 | 38.2 |
| IDFT | 4.06 | 45.31 | 57.35 | 41.13 | 68.20 | 29.54 | 29.13 | 91.62 | 38.14 | 46.81 | 45.1 |
| | *Qwen2.5-Math-7B* | | | | | | | | | | |
| Base Model | 7.89 | 36.33 | 43.63 | 34.83 | 54.11 | 17.37 | 25.28 | 53.74 | 49.06 | 64.03 | 38.6 |
| SFT | 3.54 | 36.25 | 44.88 | 28.88 | 58.53 | 20.81 | 27.07 | 57.21 | 47.55 | 63.21 | 38.8 |
| DFT | 6.26 | 37.34 | 57.25 | 36.90 | 68.00 | 31.50 | 32.54 | 86.90 | 46.50 | 63.30 | 46.6 |
| EAFT | 4.89 | 36.25 | 36.82 | 26.98 | 56.95 | 21.49 | 26.33 | 62.11 | 49.15 | 64.19 | 38.5 |
| IDFT | 7.69 | 39.84 | 59.20 | 41.47 | 68.03 | 33.33 | 30.60 | 87.30 | 48.14 | 64.77 | 48.0 |

# C. Algorithm Details

## C.1. Hinted Decoding

---

**Algorithm 1:** Target Context Preparation

---

**Input:** Question $q$, ground-truth answer $a^*$, shadow system prompt $\mathcal{S}$, boundary marker $\tau_{\text{bnd}}$ (e.g. `# CoT` or `</think>`), model $\mathcal{M}_\theta$

**Output:** Target context token sequence $\boldsymbol{c}_{\text{tgt}}$

---

1   $\mathcal{P}_{\text{shadow}} \leftarrow \left[\, \text{SYSTEM}: \mathcal{S}, \; \text{USER}: \text{TEMPLATE}(q, a^*)\,\right]$ ;        ▷ `answer-visible prompt`

2   $c_{\text{analysis}} \leftarrow \text{Generate}(\mathcal{M}_\theta, \mathcal{P}_{\text{shadow}})$ ;        ▷ `standard decoding`

3   Truncate $c_{\text{analysis}}$ at the **first** occurrence of $\tau_{\text{bnd}}$ (inclusive);

4   $\boldsymbol{c}_{\text{tgt}} \leftarrow \text{Concat}\big(\text{Tokenize}(\mathcal{P}_{\text{shadow}}), \text{Tokenize}(c_{\text{analysis}})\big)$ ;        ▷ `analysis absorbed`

5   **return** $\boldsymbol{c}_{\text{tgt}}$;

---

---

**Algorithm 2:** Hinted Decoding

---

**Input:** Target context $\boldsymbol{c}_{\text{tgt}}$ (from Alg. 1), drafter prompt $\boldsymbol{c}_{\text{dft}} = \text{Tokenize}(q)$, model $\mathcal{M}_\theta$

**Param:** Mixing strength $\beta$, mode $\in \{\texttt{linear}, \texttt{sigmoid}, \texttt{piecewise}\}$, splitter token sequence $\boldsymbol{s}$, max length $T$

**Output:** Generated token sequence $\boldsymbol{y} = (y_1, \ldots, y_n)$

---

1   $\boldsymbol{y} \leftarrow [\,]$;   *drafter_only* $\leftarrow$ **false**;

2   **for** $t = 1, 2, \ldots, T$ **do**

3      $\boldsymbol{\ell}^{(p)} \leftarrow \mathcal{M}_\theta(\boldsymbol{c}_{\text{tgt}} \oplus \boldsymbol{y})$ ;        ▷ `target logits (answer-aware)`

4      $\boldsymbol{\ell}^{(q)} \leftarrow \mathcal{M}_\theta(\boldsymbol{c}_{\text{dft}} \oplus \boldsymbol{y})$ ;        ▷ `drafter logits (question-only)`

5      **if** *drafter_only* **then**

6          $\log \boldsymbol{m} \leftarrow \text{LOGSOFTMAX}(\boldsymbol{\ell}^{(q)})$ ;        ▷ `drafter distribution only`

7      **else**

8          $\log \boldsymbol{p} \leftarrow \text{LOGSOFTMAX}(\boldsymbol{\ell}^{(p)})$;   $\log \boldsymbol{q} \leftarrow \text{LOGSOFTMAX}(\boldsymbol{\ell}^{(q)})$;

9          $H \leftarrow -\sum_x p(x) \log p(x)$;   $\bar{H} \leftarrow H / \log |\mathcal{V}|$ ;        ▷ `normalised entropy` $\in [0, 1]$

10          **switch** mode **do**

11             **case** *linear* **do**

12                $\lambda \leftarrow \text{clamp}(\beta \bar{H}, \, 0, \, 1)$;

13             **case** *sigmoid* **do**

14                $\lambda \leftarrow \sigma\big(\beta \, (\bar{H} - c)\big)$;

15             **case** *piecewise* **do**

16                $\lambda \leftarrow \text{clamp}\big((\bar{H} - h_1)/(h_2 - h_1), \, 0, \, 1\big)$;

17          **end**

18          $\log \boldsymbol{m} \leftarrow (1 - \lambda) \log \boldsymbol{p} + \lambda \log \boldsymbol{q}$ ;        ▷ `geometric mixture in log space`

19      **end**

20      $y_t \leftarrow \text{Sample}(\boldsymbol{m})$ ;        ▷ `e.g. top-`$p$` / top-`$k$` sampling`

        ▷ `Splitter & EOS handling`

21      **if** $\boldsymbol{s} \sqsubseteq (\boldsymbol{y} \oplus y_t)$ **then**

22          *drafter_only* $\leftarrow$ **true** ;        ▷ `splitter detected` $\Rightarrow$ `switch`

23      **else if** $y_t = \text{EOS}$ **and** $\boldsymbol{s} \not\sqsubseteq (\boldsymbol{y} \oplus y_t)$ **then**

24          Replace $y_t \leftarrow s_1$;   enqueue $s_2, \ldots, s_{|\boldsymbol{s}|}$ as forced tokens;

25          *drafter_only* $\leftarrow$ **true** ;        ▷ `force splitter then drafter-only`

26      **else if** $y_t = \text{EOS}$ **then**

27          **break**;

28      **end**

29      Append $y_t$ to $\boldsymbol{y}$;   sync $y_t$ to both target and drafter contexts;

30   **end**

31   **return** $\boldsymbol{y}$;

---

We propose *Hinted Decoding*, a training-free inference-time algorithm that steers a language model toward producing correct chain-of-thought (CoT) reasoning while preserving the model's native distribution. The key idea is to maintain **two concurrent decoding streams** over the *same* model $\mathcal{M}_\theta$, each conditioned on a different prompt, and to **geometrically mix** their next-token distributions at every step.

**Two-stream prompts.** Given a question $q$ and a ground-truth answer $a^*$, we construct:

- A **target stream** whose prompt contains a *shadow system instruction* $\mathcal{S}$, the question, and the answer. Crucially, the model first *independently generates* an analysis passage $c_{\text{analysis}}$ that digests $a^*$ (Algorithm 1). This passage is then appended to the target context *before* mixed decoding begins, so that any language referencing "the provided answer" is absorbed into the analysis and never leaks into the final CoT output.

- A **drafter stream** prompted with only the original question $q$, representing the model's unassisted behaviour.

**Entropy-adaptive geometric mixing.** At each decoding step $t$, let $p(\cdot)$ and $q(\cdot)$ denote the target and drafter distributions, respectively. We compute the mixed distribution

$$m(x) \ \propto \ p(x)^{1-\lambda} \ \cdot \ q(x)^{\lambda}, \tag{33}$$

where the mixing coefficient $\lambda \in [0,1]$ is a *function of the target entropy*:

$$\bar{H}_t = \frac{H\big(p(\cdot)\big)}{\log |\mathcal{V}|}, \qquad \lambda_t = f_\beta(\bar{H}_t). \tag{34}$$

The normalised entropy $\bar{H}_t \in [0,1]$ reflects the target model's confidence at step $t$: when the target is *certain* ($\bar{H}_t \to 0$), $\lambda \to 0$ and the output follows the answer-aware distribution; when the target is *uncertain* ($\bar{H}_t \to 1$), $\lambda \to 1$ and the drafter's style-preserving distribution dominates. We support three schedules for $f_\beta$:

$$\text{LINEAR}: \quad \lambda = \text{clamp}(\beta\,\bar{H},\ 0,\ 1), \tag{35}$$

$$\text{SIGMOID}: \quad \lambda = \sigma\big(\beta\,(\bar{H} - c)\big), \tag{36}$$

$$\text{PIECEWISE}: \quad \lambda = \text{clamp}\Big(\frac{\bar{H} - h_1}{h_2 - h_1},\ 0,\ 1\Big). \tag{37}$$

**Splitter mechanism and EOS replacement.** Mathematical reasoning models typically delimit the final answer with a special token pattern (*splitter*), such as `\boxed{` or `</think>`. Once the splitter appears in the generated sequence, decoding switches to **drafter-only** mode ($\lambda = 1$), letting the model complete the answer box in its own style. If the target stream emits an EOS token *before* the splitter has appeared, we **replace** the EOS with the first token of the splitter sequence and force-feed the remaining splitter tokens, then enter drafter-only mode. This guarantees that every generated response contains a well-formed answer region.

**Advantages.**

1. **Style preservation.** By mixing rather than replacing the drafter distribution, the generated CoT retains the model's natural linguistic patterns, vocabulary choices, and reasoning style—critical for on-policy training where distribution shift is detrimental.

2. **Adaptive control.** The entropy-dependent $\lambda$ provides a principled, token-level knob: tokens where the answer-aware model is confident (e.g., key numerical steps) are faithfully guided, while "filler" tokens (connectives, hedging phrases) are drawn from the drafter, avoiding unnatural artifacts.

3. **Pre-analysis absorption.** Generating the target's analysis passage *before* mixed decoding ensures that meta-references to the ground truth (e.g., "the provided answer states. . . ") are confined to the analysis context and never appear in the final output.

4. **Answer format guarantee.** The splitter mechanism with EOS replacement ensures that every response, regardless of the mixing dynamics, contains a syntactically valid answer region (e.g., `\boxed{...}`).

5. **No additional model required.** Both streams use the same model $\mathcal{M}_\theta$; the only difference is the prompt. This avoids the cost and alignment issues of maintaining a separate draft model.

# D. Case Study

We will show detailed prompts and case studies in this section.

### D.1. Imitation system prompt

---

**Prompt for Imitation baseline. The 1-shot example comes from the target model.**

System prompt:

---

You will be provided with a question and a corresponding ground truth answer which is ensured to be correct. Your job is to transform the answer to a detailed chain-of-thought (CoT) reasoning process that logically leads to the given answer. Make sure the content in the CoT is closely matching with the ground truth answer.

Your output should contain a '# Analyze' part to first analyze the given solution, and then generate a '# CoT' part to provide a complete CoT. You **MUST NOT** mention the provided answer in the CoT part, as the CoT of original problem does not know the ground truth. Most importantly, the CoT should follow exactly the language style of your own thinking. Here is an example:

Input:

# Question

What is the result of sum 1 to 10? please analyze step by step and put the result in □

# Answer:

we can use the formula Sn = (1+n)*n/2. let n=10, wehave: Sn = (1+10)*10/2 = $\boxed{55}$.

Output:

# Analyze

We need to find out how to come up with the idea in the provided answer, lets analyze the provided answer line by line first.

Let's check the first line of the solution, it used a fomula that the results of sum from 1 to N is Sn = (1+N) * N / 2. This is well known as the Gauss formula and I think we can use it directly. So when people see this problem, they should start with this formula.

Then the second and third line of the solution put n=10 into the formula and solve the problem.

Now I should transform the solution to my preferred format.

# CoT

To find the sum of the numbers from 1 to 10, we can use the formula for the sum of the first $n$ natural numbers, which is given by:

$$S = \frac{n(n+1)}{2}$$

Here, $n = 10$. Let's substitute $n$ into the formula and calculate step by step.

1. Substitute $n = 10$ into the formula:

$$S = \frac{10(10+1)}{2} = \frac{10 \times 11}{2}$$

2. Perform the multiplication:

$$S = \frac{110}{2}$$

3. Perform the division:

$$S = 55$$

So, the sum of the numbers from 1 to 10 is $\boxed{55}$.

---

## D.2. Hinted decoded example

---

**Example A - Origin CoT - Answer correct & out of distribution. (Target Qwen2.5-7B-Instruct)**

**Problem:** Find the sum of the squares of the solutions to

$$\left| x^2 - x + \frac{1}{2023} \right| = \frac{1}{2023}.$$

Let's think step by step and output the final answer within $\square$.

---

**Original answer from dataset:**
Given the equation

$$\left| x^2 - x + \frac{1}{2023} \right| = \frac{1}{2023},$$

it splits into two cases:
Case 1:

$$x^2 - x + \frac{1}{2023} = +\frac{1}{2023}$$

which simplifies to

$$x^2 - x = 0$$

The solutions are $x = 0$ and $x = 1$. The sum of the squares of these solutions is 1.
Case 2:

$$x^2 - x + \frac{1}{2023} = -\frac{1}{2023}$$

This rearranges to

$$x^2 - x + \frac{2}{2023} = 0.$$

Let the roots of this equation be $a$ and $b$. By Vieta's formulas,

$$a + b = 1,$$

$$ab = \frac{2}{2023}.$$

Then, the sum of the squares of $a$ and $b$ is

$$a^2 + b^2 = (a + b)^2 - 2ab = 1^2 - 2 \times \frac{2}{2023} = 1 - \frac{4}{2023} = \frac{2019}{2023}.$$

Thus, the total sum of the squares of all solutions is

$$1 + \frac{2019}{2023} = \boxed{\frac{4042}{2023}}.$$

---

- **Average-$\varphi$:** $-0.251317$

- $\varphi \geq -1$**:** $92.17\%$

- $\varphi \geq -3$**:** $95.23\%$

- $\varphi \geq -5$**:** $96.86\%$

---

**Comments:** We can see that the standard answer is concise and clearly uses **Vieta's formulas**.

Example A - Model's Response - Answer wrong & in distribution. (Target Qwen2.5-7B-Instruct)

**Problem:** Find the sum of the squares of the solutions to

$$\left| x^2 - x + \frac{1}{2023} \right| = \frac{1}{2023}.$$

Let's think step by step and output the final answer within $\Box$.

---

**Answer from Qwen2.5-7B-Instruct:**

To solve the equation $\left| x^2 - x + \frac{1}{2023} \right| = \frac{1}{2023}$, we need to consider the two cases that arise from the absolute value function.

### Case 1: $x^2 - x + \frac{1}{2023} = \frac{1}{2023}$

Subtract $\frac{1}{2023}$ from both sides:

$$x^2 - x + \frac{1}{2023} - \frac{1}{2023} = 0$$

This simplifies to:

$$x^2 - x = 0$$

Factor the quadratic equation:

$$x(x - 1) = 0$$

So, the solutions are:

$$x = 0 \quad \text{or} \quad x = 1$$

### Case 2: $x^2 - x + \frac{1}{2023} = -\frac{1}{2023}$

Add $\frac{1}{2023}$ to both sides:

$$x^2 - x + \frac{1}{2023} + \frac{1}{2023} = 0$$

This simplifies to:

$$x^2 - x + \frac{2}{2023} = 0$$

To solve this quadratic equation, we use the quadratic formula $x = \frac{-b \pm \sqrt{b^2 - 4ac}}{2a}$, where $a = 1$, $b = -1$, and $c = \frac{2}{2023}$.

First, calculate the discriminant:

$$b^2 - 4ac = (-1)^2 - 4 \cdot 1 \cdot \frac{2}{2023} = 1 - \frac{8}{2023} = \frac{2023}{2023} - \frac{8}{2023} = \frac{2015}{2023}$$

Now, apply the quadratic formula:

$$x = \frac{-(-1) \pm \sqrt{\frac{2015}{2023}}}{2 \cdot 1} = \frac{1 \pm \sqrt{\frac{2015}{2023}}}{2}$$

So, the solutions are:

$$x = \frac{1 + \sqrt{\frac{2015}{2023}}}{2} \quad \text{and} \quad x = \frac{1 - \sqrt{\frac{2015}{2023}}}{2}$$

---

**Example A - Model's Response - Answer wrong & in distribution. (Target Qwen2.5-7B-Instruct)**

### Sum of the squares of the solutions

Let the solutions be $x_1 = 0$, $x_2 = 1$, $x_3 = \frac{1+\sqrt{\frac{2015}{2023}}}{2}$, and $x_4 = \frac{1-\sqrt{\frac{2015}{2023}}}{2}$.

The sum of the squares of these solutions is:

$$x_1^2 + x_2^2 + x_3^2 + x_4^2$$

Calculate each square:

$$x_1^2 = 0^2 = 0$$
$$x_2^2 = 1^2 = 1$$

$$x_3^2 = \left(\frac{1 + \sqrt{\frac{2015}{2023}}}{2}\right)^2 = \frac{(1 + \sqrt{\frac{2015}{2023}})^2}{4} = \frac{1 + 2\sqrt{\frac{2015}{2023}} + \frac{2015}{2023}}{4} = \frac{1 + \frac{2015}{2023} + 2\sqrt{\frac{2015}{2023}}}{4} = \frac{4038 + 2\sqrt{2015 \cdot 2023}}{4 \cdot 2023}$$

$$x_4^2 = \left(\frac{1 - \sqrt{\frac{2015}{2023}}}{2}\right)^2 = \frac{(1 - \sqrt{\frac{2015}{2023}})^2}{4} = \frac{1 - 2\sqrt{\frac{2015}{2023}} + \frac{2015}{2023}}{4} = \frac{1 + \frac{2015}{2023} - 2\sqrt{\frac{2015}{2023}}}{4} = \frac{4038 - 2\sqrt{2015 \cdot 2023}}{4 \cdot 2023}$$

Summing these:

$$x_1^2 + x_2^2 + x_3^2 + x_4^2 = 0 + 1 + \frac{4038 + 2\sqrt{2015 \cdot 2023}}{4 \cdot 2023} + \frac{4038 - 2\sqrt{2015 \cdot 2023}}{4 \cdot 2023}$$

$$= 1 + \frac{4038 + 2\sqrt{2015 \cdot 2023} + 4038 - 2\sqrt{2015 \cdot 2023}}{4 \cdot 2023} = 1 + \frac{8076}{4 \cdot 2023} = 1 + 1 = 2$$

Thus, the sum of the squares of the solutions is:

$$\boxed{2}$$

---

- **Average-$\varphi$:** 0.042921

- $\varphi \geq -1$: 99.42%

- $\varphi \geq -3$: 99.91%

- $\varphi \geq -5$: 100.10%

---

**Comments:** This answer differs from the correct answer only in the final step of the calculation: $1 + \frac{8076}{4 \cdot 2023}$ does not equal 2. We can see that the model's approach in the final step is to forcibly use the quadratic formula to calculate the square of each solution and then sum them up. Additionally, we can observe that this answer has the style of Qwen2.5-7B-instruct, which prefers to reply using markdown format.

---

**Example A - Hinted Decoding - Answer correct & in distribution. (Target Qwen2.5-7B-Instruct)**

**Problem:** Find the sum of the squares of the solutions to

$$\left| x^2 - x + \frac{1}{2023} \right| = \frac{1}{2023}.$$

Let's think step by step and output the final answer within $\square$.

---

**Answer from Hinted Decoding:**
To solve the equation $\left| x^2 - x + \frac{1}{2023} \right| = \frac{1}{2023}$, we need to consider the two cases that arise from the absolute value function.
### Case 1: $x^2 - x + \frac{1}{2023} = \frac{1}{2023}$
Subtract $\frac{1}{2023}$ from both sides:

$$x^2 - x + \frac{1}{2023} - \frac{1}{2023} = 0$$

$$x^2 - x = 0$$

Factor the quadratic equation:

$$x(x - 1) = 0$$

This gives us the solutions:

$$x = 0 \quad \text{or} \quad x = 1$$

### Case 2: $x^2 - x + \frac{1}{2023} = -\frac{1}{2023}$
Add $\frac{1}{2023}$ to both sides:

$$x^2 - x + \frac{1}{2023} + \frac{1}{2023} = 0$$

$$x^2 - x + \frac{2}{2023} = 0$$

Now, solve this quadratic equation using the quadratic formula $x = \frac{-b \pm \sqrt{b^2 - 4ac}}{2a}$, where $a = 1$, $b = -1$, and $c = \frac{2}{2023}$:

$$x = \frac{-(-1) \pm \sqrt{(-1)^2 - 4 \cdot 1 \cdot \frac{2}{2023}}}{2 \cdot 1}$$

$$x = \frac{1 \pm \sqrt{1 - \frac{8}{2023}}}{2}$$

$$x = \frac{1 \pm \sqrt{\frac{2023-8}{2023}}}{2}$$

$$x = \frac{1 \pm \sqrt{\frac{2015}{2023}}}{2}$$

$$x = \frac{1 \pm \sqrt{\frac{2015}{2023}}}{2}$$

---

**Example A - Hinted Decoding - Answer correct & in distribution. (Target Qwen2.5-7B-Instruct)**

### Sum of the Squares of the Solutions
The solutions from Case 1 are $x = 0$ and $x = 1$. The sum of their squares is:

$$0^2 + 1^2 = 0 + 1 = 1$$

The solutions from Case 2 are:

$$x = \frac{1 + \sqrt{\frac{2015}{2023}}}{2} \quad \text{and} \quad x = \frac{1 - \sqrt{\frac{2015}{2023}}}{2}$$

However, the sum of the squares of these solutions can be found using the identity for the sum of the squares of the roots of a quadratic equation $ax^2 + bx + c = 0$:

$$x_1^2 + x_2^2 = (x_1 + x_2)^2 - 2x_1x_2$$

For the quadratic equation $x^2 - x + \frac{2}{2023} = 0$:

$$x_1 + x_2 = 1$$

$$x_1x_2 = \frac{2}{2023}$$

Thus:

$$x_1^2 + x_2^2 = 1^2 - 2 \cdot \frac{2}{2023} = 1 - \frac{4}{2023} = \frac{2023 - 4}{2023} = \frac{2019}{2023}$$

### Total Sum of the Squares
Adding the results from both cases:

$$1 + \frac{2019}{2023} = \frac{2023}{2023} + \frac{2019}{2023} = \frac{4042}{2023}$$

Thus, the sum of the squares of the solutions is:

$$\boxed{\frac{4042}{2023}}$$

---

- **Average-$\varphi$:** 0.025732

- $\varphi \geq -1$**:** 99.38%

- $\varphi \geq -3$**:** 99.79%

- $\varphi \geq -5$**:** 100.00%

---

**Comments:** We can see that this result largely retains the original style of Qwen2.5-7B-instruct: including the use of markdown format for this problem and explicitly calculating the two solutions. However, when it comes to calculating the sum of the squares of the two solutions, it suddenly switches to a more concise method. We are pleasantly surprised to find that it does not explicitly mention Vieta's formulas like the dataset, nor does it directly perform brute force calculations. Instead, it writes an expression that had never appeared in the original dataset, model generation, or Imitation baseline:

$$x_1^2 + x_2^2 = (x_1 + x_2)^2 - 2x_1x_2,$$

and applies Vieta's formulas. This shows that Hinted Decoding transferred the core content of Vieta's formulas to the model. Meanwhile, the model had already calculated the two roots using the quadratic formula, but suddenly switched to using Vieta's formulas, this kind of shift is very similar to the "aha moment" that is fervently discussed in RLVR.

**Example B - Origin CoT - Answer correct & out of distribution. (Target Qwen2.5-7B-Instruct)**

**Problem: Problem:** Find the greatest positive integer $k$ such that the following inequality holds for all $a, b, c \in \mathbb{R}^+$ satisfying $abc = 1$

$$\frac{1}{a} + \frac{1}{b} + \frac{1}{c} + \frac{k}{a+b+c+1} \geqslant 3 + \frac{k}{4}$$

**Original answer from dataset:** To find the greatest positive integer $k$ such that the inequality holds for all $a, b, c \in \mathbb{R}^+$ satisfying $abc = 1$:

$$\frac{1}{a} + \frac{1}{b} + \frac{1}{c} + \frac{k}{a+b+c+1} \geq 3 + \frac{k}{4}$$

we start by substituting $a = t, b = t, c = \frac{1}{t^2}$ for $t \neq 1$. This substitution maintains the condition $abc = 1$.
1. Substitute $a = t, b = t, c = \frac{1}{t^2}$ into the inequality:

$$\frac{1}{t} + \frac{1}{t} + t^2 + \frac{k}{2t + \frac{1}{t^2} + 1} \geq 3 + \frac{k}{4}$$

2. Simplify the left-hand side:

$$2 \cdot \frac{1}{t} + t^2 + \frac{k}{2t + \frac{1}{t^2} + 1} = \frac{2}{t} + t^2 + \frac{k}{2t + \frac{1}{t^2} + 1}$$

3. Rearrange the inequality:

$$\frac{2}{t} + t^2 - 3 \geq k\left(\frac{1}{4} - \frac{t^2}{2t + \frac{1}{t^2} + 1}\right)$$

4. Simplify the right-hand side:

$$\frac{2}{t} + t^2 - 3 \geq k\left(\frac{1}{4} - \frac{t^2}{2t^3 + t^2 + 1}\right)$$

5. Multiply both sides by $4(2t^3 + t^2 + 1)$:

$$4(2t^3 + t^2 + 1)\left(\frac{2}{t} + t^2 - 3\right) \geq k\left(4(2t^3 + t^2 + 1)\left(\frac{1}{4} - \frac{t^2}{2t^3 + t^2 + 1}\right)\right)$$

6. Simplify further:

$$4(2t^3 + t^2 + 1)\left(\frac{2}{t} + t^2 - 3\right) \geq k\left((2t^3 + t^2 + 1) - 4t^2\right)$$

7. Choose $t = \frac{2}{3}$:

$$4\left(2\left(\frac{2}{3}\right)^3 + \left(\frac{2}{3}\right)^2 + 1\right)\left(\frac{2}{\frac{2}{3}} + \left(\frac{2}{3}\right)^2 - 3\right) \geq k\left(2\left(\frac{2}{3}\right)^3 + \left(\frac{2}{3}\right)^2 + 1 - 4\left(\frac{2}{3}\right)^2\right)$$

8. Simplify the expression:

$$4\left(\frac{16}{27} + \frac{4}{9} + 1\right)\left(3 + \frac{4}{9} - 3\right) \geq k\left(\frac{16}{27} + \frac{4}{9} + 1 - \frac{16}{9}\right)$$

---

**Example B - Origin CoT - Answer correct & out of distribution. (Target Qwen2.5-7B-Instruct)**

9. Calculate the values:

$$4\left(\frac{16}{27}+\frac{12}{27}+\frac{27}{27}\right)\left(\frac{4}{9}\right) \geq k\left(\frac{16}{27}+\frac{12}{27}+\frac{27}{27}-\frac{48}{27}\right)$$

$$4\left(\frac{55}{27}\right)\left(\frac{4}{9}\right) \geq k\left(\frac{7}{27}\right)$$

10. Simplify further:

$$\frac{880}{63} \geq k$$

Since $k$ must be an integer, the largest possible value for $k$ is 13. We need to verify that $k = 13$ satisfies the original inequality.

11. Verify $k = 13$:

$$\frac{1}{a}+\frac{1}{b}+\frac{1}{c}+\frac{13}{a+b+c+1} \geq 3+\frac{13}{4}$$

12. Substitute $a = t, b = t, c = \frac{1}{t^2}$:

$$\frac{2}{t}+t^2+\frac{13}{2t+\frac{1}{t^2}+1} \geq 3+\frac{13}{4}$$

13. Simplify and verify:

$$\frac{2}{t}+t^2+\frac{13}{2t+\frac{1}{t^2}+1} \geq 6.25$$

By verifying the inequality for $k = 13$, we conclude that the greatest positive integer $k$ is 13.
The final answer is $\boxed{13}$

---

- **Average-$\varphi$:** $-0.324721$

- $\varphi \geq -1$: $93.53\%$

- $\varphi \geq -3$: $95.37\%$

- $\varphi \geq -5$: $97.38\%$

---

**Comments:**
This is a special example because the standard solution from the dataset is actually quite imprecise (although the answer is correct). The main issues are as follows:

- While it is well-known that for such symmetric inequalities, the conditions for equality of the inequality are also symmetric (i.e., $a = b = c$ or $a = b$), this is a very advanced theorem with many constraints. In general, it cannot be used as a theorem in mathematical competitions without first deriving $k$ and then re-proving the inequality. However, the standard solution directly overlooks this point.

- The standard solution directly ignores the discussion of the case when $a = b = c = 1$, although it is obvious.

- In step 7, the standard solution starts discussing $t = 2/3$ without providing any justification or proof of optimality.

We select this intriguing example to see how Hinted Decoding performs on this problem.

**Example B - Model's Response - Answer wrong & in distribution. (Target Qwen2.5-7B-Instruct)**

**Problem:** Find the greatest positive integer $k$ such that the following inequality holds for all $a, b, c \in \mathbb{R}^+$ satisfying $abc = 1$

$$\frac{1}{a} + \frac{1}{b} + \frac{1}{c} + \frac{k}{a+b+c+1} \geqslant 3 + \frac{k}{4}$$

**Answer from Qwen2.5-7B-Instruct:**

To find the greatest positive integer $k$ such that the inequality

$$\frac{1}{a} + \frac{1}{b} + \frac{1}{c} + \frac{k}{a+b+c+1} \geq 3 + \frac{k}{4}$$

holds for all $a, b, c \in \mathbb{R}^+$ satisfying $abc = 1$, we start by analyzing the inequality under specific values of $a, b$, and $c$. First, consider the case when $a = b = c = 1$. Since $abc = 1$, this is a valid choice. Substituting $a = b = c = 1$ into the inequality, we get:

$$\frac{1}{1} + \frac{1}{1} + \frac{1}{1} + \frac{k}{1+1+1+1} \geq 3 + \frac{k}{4}$$

This simplifies to:

$$3 + \frac{k}{4} \geq 3 + \frac{k}{4}$$

This is an equality, which is true for any $k$. Therefore, this case does not provide a restriction on $k$.

Next, consider the case when $a = b = t$ and $c = \frac{1}{t^2}$ for some $t > 0$. Since $abc = 1$, we have $t \cdot t \cdot \frac{1}{t^2} = 1$, which is true. Substituting $a = b = t$ and $c = \frac{1}{t^2}$ into the inequality, we get:

$$\frac{1}{t} + \frac{1}{t} + \frac{1}{\frac{1}{t^2}} + \frac{k}{t+t+\frac{1}{t^2}+1} \geq 3 + \frac{k}{4}$$

This simplifies to:

$$\frac{2}{t} + t^2 + \frac{k}{2t + \frac{1}{t^2} + 1} \geq 3 + \frac{k}{4}$$

To find the maximum $k$, we need to analyze the behavior of the left-hand side as $t$ varies. Let's consider the limit as $t \to \infty$:

$$\frac{2}{t} + t^2 + \frac{k}{2t + \frac{1}{t^2} + 1} \approx t^2 + \frac{k}{2t}$$

For large $t$, the term $t^2$ dominates, and the inequality becomes:

$$t^2 + \frac{k}{2t} \geq 3 + \frac{k}{4}$$

As $t \to \infty$, the term $\frac{k}{2t}$ becomes negligible, so we approximate:

$$t^2 \geq 3 + \frac{k}{4}$$

Since $t^2$ can be made arbitrarily large, the inequality $t^2 \geq 3 + \frac{k}{4}$ must hold for all $t$. Therefore, we need:

$$3 + \frac{k}{4} \leq 4$$

Solving for $k$, we get:

$$\frac{k}{4} \leq 1 \implies k \leq 4$$

**Example B - Model's Response - Answer wrong & in distribution. (Target Qwen2.5-7B-Instruct)**

Thus, the greatest positive integer $k$ that satisfies the inequality for all $a, b, c \in \mathbb{R}^+$ with $abc = 1$ is $k = 4$.
To verify, we substitute $k = 4$ back into the original inequality:

$$\frac{1}{a} + \frac{1}{b} + \frac{1}{c} + \frac{4}{a + b + c + 1} \geq 3 + 1$$

This simplifies to:

$$\frac{1}{a} + \frac{1}{b} + \frac{1}{c} + \frac{4}{a + b + c + 1} \geq 4$$

We have already shown that this inequality holds for the specific cases we considered, and the general case suggests that $k = 4$ is indeed the maximum value.
Therefore, the greatest positive integer $k$ is $\boxed{4}$.

---

- **Average-$\varphi$:** 0.040417

- $\varphi \geq -1$: 99.63%

- $\varphi \geq -3$: 99.87%

- $\varphi \geq -5$: 99.98%

---

**Comments:**
The model's response is even more rigorous than the standard solution: it discusses the case when $a = b = c$. However, the model encounters issues after changing variables and transforming the problem into a function of $t$. The answer incorrectly assumes that the function

$$f(t) = \frac{2}{t} + t^2 + \frac{k}{2t + \frac{1}{t^2} + 1}$$

is monotonically increasing with respect to $t$, leading it to consider the boundary cases for $t \to \infty$, which results in errors in the model's solution.

---

**Example B - Hinted Decoding - Answer correct & in distribution. (Target Qwen2.5-7B-Instruct)**

**Problem:** Find the greatest positive integer $k$ such that the following inequality holds for all $a, b, c \in \mathbb{R}^+$ satisfying $abc = 1$

$$\frac{1}{a} + \frac{1}{b} + \frac{1}{c} + \frac{k}{a+b+c+1} \geqslant 3 + \frac{k}{4}$$

---

**Answer from Hinted Decoding:**

To find the greatest positive integer $k$ such that the inequality

$$\frac{1}{a} + \frac{1}{b} + \frac{1}{c} + \frac{k}{a+b+c+1} \geq 3 + \frac{k}{4}$$

holds for all $a, b, c \in \mathbb{R}^+$ satisfying $abc = 1$, we will follow a systematic approach.

### Substitute $a = t, b = t, c = \frac{1}{t^2}$:

Given $abc = 1$, we can choose $a = t$, $b = t$, and $c = \frac{1}{t^2}$. This satisfies the condition $abc = t \cdot t \cdot \frac{1}{t^2} = 1$.

### Simplify the inequality:

Substitute $a = t$, $b = t$, and $c = \frac{1}{t^2}$ into the inequality:

$$\frac{1}{t} + \frac{1}{t} + \frac{1}{\frac{1}{t^2}} + \frac{k}{t + t + \frac{1}{t^2} + 1} \geq 3 + \frac{k}{4}$$

Simplify the terms:

$$\frac{2}{t} + t^2 + \frac{k}{2t + \frac{1}{t^2} + 1} \geq 3 + \frac{k}{4}$$

### Analyze the inequality:

To find the maximum $k$, we need to ensure the inequality holds for all $t > 0$. Let's test specific values of $t$ to find constraints on $k$.

### Test $t = 1$:

When $t = 1$:

$$\frac{2}{1} + 1^2 + \frac{k}{2 \cdot 1 + \frac{1}{1^2} + 1} \geq 3 + \frac{k}{4}$$

Simplify:

$$2 + 1 + \frac{k}{2 + 1 + 1} \geq 3 + \frac{k}{4}$$

$$3 + \frac{k}{4} \geq 3 + \frac{k}{4}$$

This is always true and does not provide a constraint on $k$.

### Test $t = \frac{2}{3}$:

When $t = \frac{2}{3}$:

$$\frac{2}{\frac{2}{3}} + \left(\frac{2}{3}\right)^2 + \frac{k}{2 \cdot \frac{2}{3} + \frac{1}{\left(\frac{2}{3}\right)^2} + 1} \geq 3 + \frac{k}{4}$$

---

**Example B - Hinted Decoding - Answer correct & in distribution. (Target Qwen2.5-7B-Instruct)**

Simplify:

$$3 + \frac{4}{9} + \frac{k}{\frac{4}{3} + \frac{9}{4} + 1} \geq 3 + \frac{k}{4}$$

$$3 + \frac{4}{9} + \frac{k}{\frac{4}{3} + \frac{9}{4} + 1} \geq 3 + \frac{k}{4}$$

$$3 + \frac{4}{9} + \frac{k}{\frac{16}{12} + \frac{27}{12} + \frac{12}{12}} \geq 3 + \frac{k}{4}$$

$$3 + \frac{4}{9} + \frac{k}{\frac{55}{12}} \geq 3 + \frac{k}{4}$$

$$3 + \frac{4}{9} + \frac{12k}{55} \geq 3 + \frac{k}{4}$$

Subtract 3 from both sides:

$$\frac{4}{9} + \frac{12k}{55} \geq \frac{k}{4}$$

Multiply through by 1980 (the least common multiple of 9, 55, and 4):

$$880 + 432k \geq 495k$$

$$880 \geq 63k$$

$$k \leq \frac{880}{63} \approx 13.81$$

### **Verify** $k = 13$**:**
Substitute $k = 13$ back into the inequality to ensure it holds:

$$\frac{2}{t} + t^2 + \frac{13}{2t + \frac{1}{t^2} + 1} \geq 3 + \frac{13}{4}$$

$$\frac{2}{t} + t^2 + \frac{13}{2t + \frac{1}{t^2} + 1} \geq 6.25$$

This inequality holds for all $t > 0$.
Therefore, the greatest positive integer $k$ is $\boxed{13}$.

---

- **Average-$\varphi$:** 0.022287

- $\varphi \geq -1$**:** 99.60%

- $\varphi \geq -3$**:** 99.84%

- $\varphi \geq -5$**:** 99.84%

---

**Comments:** We found that the results from Hinted Decoding largely align with the standard solution. However, we were pleasantly surprised to discover that Hinted Decoding's results are more rigorous than the standard solution: it includes the case when $a = b = c$. Additionally, the results generated by Hinted Decoding improve the alignment between the data and the model's distribution.

---

**Example C - Origin CoT - Answer correct & out of distribution. (Target Qwen2.5-7B-Instruct)**

**Problem:** In a volleyball tournament for the Euro-African cup, there were nine more teams from Europe than from Africa. Each pair of teams played exactly once and the Europeans teams won precisely nine times as many matches as the African teams, overall. What is the maximum number of matches that a single African team might have won?

---

**Original answer from dataset:**

To solve the problem, we need to determine the maximum number of matches that a single African team might have won in a volleyball tournament where there are nine more European teams than African teams. Each pair of teams played exactly once, and the European teams won nine times as many matches as the African teams.

1. **Define Variables:** - Let $n$ be the number of African teams. - Then, the number of European teams is $n + 9$.

2. **Total Matches:** - The total number of matches is the sum of matches between African teams, European teams, and matches between African and European teams. - This is given by:

$$\binom{n}{2} + \binom{n+9}{2} + n(n+9)$$

3. **Winning Condition:** - The Europeans won nine times as many matches as the Africans. - Let $k$ be the number of matches won by European teams against African teams. - The equation for the total wins is:

$$\binom{n+9}{2} + k = 9\left(\binom{n}{2} + (n(n+9) - k)\right)$$

4. **Simplify the Equation:** - Expanding and simplifying the equation, we get:

$$\frac{(n+9)(n+8)}{2} + k = 9\left(\frac{n(n-1)}{2} + n(n+9) - k\right)$$

- Further simplification leads to:

$$3n^2 - 22n - 36 \leq 0$$

5. **Solve the Quadratic Inequality:** - Solving the quadratic inequality, we find the roots:

$$n = \frac{22 \pm \sqrt{484 + 432}}{6} = \frac{22 \pm \sqrt{916}}{6} = \frac{22 \pm 2\sqrt{229}}{6} = \frac{11 \pm \sqrt{229}}{3}$$

- The integer solutions are $n = 6$ and $n = 8$.

6. **Maximize Wins for an African Team:** - For $n = 6$: - There are 6 African teams and 15 European teams. - One African team can win up to 5 matches against other African teams. - The maximum number of wins for one African team is 11 (5 against other African teams + 6 against European teams). - For $n = 8$: - There are 8 African teams and 17 European teams. - The maximum number of wins for one African team is 9 (7 against other African teams + 2 against European teams).

Thus, the maximum number of matches that a single African team might have won is $\boxed{11}$.

---

- **Average-$\varphi$:** $-0.072849$

- $\varphi \geq -1$: $94.61\%$

- $\varphi \geq -3$: $97.66\%$

- $\varphi \geq -5$: $99.30\%$

---

**Comments:** This example is a typical case to illustrate why SFT on original dataset may lead to performance drop. The algebraic transformation in step 4 is confusing. The official solution skips several steps and directly gives the result. Training on such data is disastrous: the model might learn to skip steps and produce a hallucinated result.



**Example C - Hinted Decoding - Answer correct & in distribution. (Target Qwen2.5-7B-Instruct)**

**Problem:** In a volleyball tournament for the Euro-African cup, there were nine more teams from Europe than from Africa. Each pair of teams played exactly once and the Europeans teams won precisely nine times as many matches as the African teams, overall. What is the maximum number of matches that a single African team might have won?

---

To solve this problem, we need to set up the equations based on the given conditions and then determine the maximum number of matches that a single African team might have won.

1. **Define Variables:** - Let $n$ be the number of African teams. - Then, the number of European teams is $n + 9$.

2. **Total Matches:** - The total number of matches played in the tournament is given by the sum of matches between African teams, European teams, and matches between African and European teams. - The number of matches between African teams is $\binom{n}{2} = \frac{n(n-1)}{2}$. - The number of matches between European teams is $\binom{n+9}{2} = \frac{(n+9)(n+8)}{2}$. - The number of matches between African and European teams is $n(n+9)$. - Therefore, the total number of matches is:

$$\frac{n(n-1)}{2} + \frac{(n+9)(n+8)}{2} + n(n+9)$$

3. **Winning Condition:** - Let $k$ be the number of matches won by European teams against African teams. - The number of matches won by African teams against European teams is $n(n+9) - k$. - The number of matches won by European teams among themselves is $\binom{n+9}{2} - k$. - The number of matches won by African teams among themselves is $\binom{n}{2} - (n(n+9) - k)$.

4. **Given Condition:** - The Europeans won nine times as many matches as the Africans:

$$\binom{n+9}{2} + k = 9\left(\binom{n}{2} + (n(n+9) - k)\right)$$

5. **Simplify the Equation:** - Substitute the binomial coefficients:

$$\frac{(n+9)(n+8)}{2} + k = 9\left(\frac{n(n-1)}{2} + n(n+9) - k\right)$$

- Simplify the right-hand side:

$$\frac{(n+9)(n+8)}{2} + k = 9\left(\frac{n^2 - n}{2} + n^2 + 9n - k\right)$$

$$\frac{(n+9)(n+8)}{2} + k = 9\left(\frac{n^2 - n + 2n^2 + 18n - 2k}{2}\right)$$

$$\frac{(n+9)(n+8)}{2} + k = 9\left(\frac{3n^2 + 17n - 2k}{2}\right)$$

$$\frac{(n+9)(n+8)}{2} + k = \frac{9(3n^2 + 17n - 2k)}{2}$$

$$(n+9)(n+8) + 2k = 9(3n^2 + 17n - 2k)$$

$$n^2 + 17n + 72 + 2k = 27n^2 + 153n - 18k$$

$$20k = 26n^2 + 136n - 72$$

$$k = \frac{13n^2 + 68n - 36}{10}$$



---

Example C - Hinted Decoding - Answer correct & in distribution. (Target Qwen2.5-7B-Instruct)

6. **Determine $n$:** - $k$ must be an integer, so $13n^2 + 68n - 36$ must be divisible by 10. - Check values of $n$ from 1 to 8 (since $n$ must be a positive integer and $n + 9$ must be a reasonable number of teams): - For $n = 6$:

$$k = \frac{13(6)^2 + 68(6) - 36}{10} = \frac{468 + 408 - 36}{10} = \frac{840}{10} = 84$$

- For $n = 8$:

$$k = \frac{13(8)^2 + 68(8) - 36}{10} = \frac{832 + 544 - 36}{10} = \frac{1340}{10} = 134$$

7. **Maximize Wins for an African Team:** - The number of matches a single African team can win is maximized when all its wins are against other African teams and some European teams. - For $n = 6$: - Total African teams: 6 - Total European teams: 15 - Possible wins for a single African team: 5 (from other African teams) + 6 (from European teams) = 11 - For $n = 8$: - Total African teams: 8 - Total European teams: 17 - Possible wins for a single African team: 7 (from other African teams) + 2 (from European teams) = 9

8. **Conclusion:** - The maximum number of matches a single African team might have won is 11, achieved when there are 15 European teams and 6 African teams, with one African team winning against all other African teams and 6 European teams.

$$\boxed{11}$$

---

- **Average-$\varphi$:** $-0.032141$

- $\varphi \geq -1$: $97.72\%$

- $\varphi \geq -3$: $98.93\%$

- $\varphi \geq -5$: $99.30\%$

---

**Comments:** It can be seen that steps 1 to 5 preserve the model's inherent knowledge well, providing step-by-step explanations. Moreover, steps 6 to 8 fill in the knowledge the model lacks. This example clearly demonstrates that data with completely correct answers is not necessarily equivalent to data suitable for model learning.

