# OpenReview forum: "Towards On-Policy SFT: Distribution Discriminant Theory and its Applications in LLM Training"
_ICML.cc/2026/Conference — ICML 2026 regular_

### Official Review · Reviewer_GKNj · 2026-03-11

**Soundness:** 3
**Presentation:** 2
**Significance:** 3
**Originality:** 3
**Overall Recommendation:** 3
**Confidence:** 4

**Summary:**

This paper addresses the generalization gap between Supervised Fine-Tuning (SFT) and Reinforcement Learning (RL) in the post-training stage of Large Language Models (LLMs). The authors argue that SFT's inferior generalization stems from forcing the model to fit external out-of-distribution data equally, which can damage pre-trained knowledge and cause catastrophic forgetting. To bridge this gap, the authors introduce Distribution Discriminant Theory (DDT), which uses a Centered Log-Likelihood (CLL) criterion to quantify whether data aligns with a model's native distribution. Building on DDT, the paper proposes two applications to enable "On-Policy SFT": In-Distribution Fine-Tuning (IDFT), a dynamic loss-level modulation that reweights the SFT objective to suppress gradients for out-of-distribution noise and amplify them for high-confidence logic ; and Hinted Decoding, a data-level technique that rewrites training data to align with the model's inherent distribution while maintaining ground-truth correctness. Extensive experiments on mathematical reasoning and coding benchmarks indicate that this framework matches the performance of offline RL algorithms, such as DPO and SimPO, while maintaining the computational efficiency of an SFT pipeline.

**Compliance With Llm Reviewing Policy:**

Affirmed.

**Final Justification:**

Based on the author's rebuttal, I increased my score.

**Key Questions For Authors:**

* To truly prove the effectiveness of the proposed method, can you provide experimental results on high-quality training data where standard SFT actually yields a significant positive gain? A valid demonstration of superiority requires showing that the proposed method can learn more effectively than baselines, rather than simply competing to see which method degrades the least when exposed to poor data.
* How would the proposed framework perform when compared directly against online reinforcement learning baselines like PPO or GRPO?
* How does the IDFT objective perform on open-ended, subjective, or creative writing tasks where "ground truth" correctness is more ambiguous than in mathematics or code?
* Given the inverted-U performance curve observed with the truncation threshold $\tau$, how can practitioners determine the optimal learning interval for entirely new domains without relying on extensive grid searches?

**Limitations:**

yes

**Strengths And Weaknesses:**

### Strengths

*
**Strong Theoretical Foundation:** The paper provides a rigorous statistical framework grounded in Signal Detection Theory to justify using Centered Log-Likelihood (CLL) over standard raw log-likelihood. The authors mathematically prove that CLL strictly maximizes the Signal-to-Noise Ratio (SNR) by eliminating context difficulty variance.


*
**Efficiency and Performance Parity:** The proposed methods achieve generalization on par with well-known offline RL algorithms (such as DPO, RPO, and SimPO) but require a lower computational budget, demonstrating high data efficiency.


*
**Mitigation of Catastrophic Forgetting:** IDFT successfully addresses a major flaw in standard SFT by acting as a selective filter that prevents the model from fitting to noise. Experiments show it maintains or surpasses baseline performance on general reasoning benchmarks like MMLU and ARC-Challenge.



---

### Limitations & Weaknesses

*
**Absence of Online RL Baselines:** The paper only compares against offline RL methods such as Reject Sampling, DPO, and SimPO. Given that current state-of-the-art paradigms rely heavily on online RL algorithms like PPO or GRPO, omitting these comparisons makes the core claim of "bridging the gap to RL" difficult to fully substantiate.


*
**Dependency on Ground Truth for Hinted Decoding:** The proposed "Hinted Decoding" technique requires the ground truth answer as an input to the imitation prompt. This fundamentally restricts the method to being a data-rewriting strategy for existing, fully labeled supervised datasets. Unlike true RL methods that can explore and learn from scalar rewards without knowing the exact target trajectory, Hinted Decoding cannot operate in exploratory environments, which slightly overstates its equivalence to RL.


*
**Confounding Variables in Data Quality ("Less is More" Paradox):** The experimental results show that standard SFT causes an average performance drop of over 10 points compared to the original un-trained instruct models (e.g., Qwen2.5-7B-instruct drops from 49.6 to 32.4 , and Phi4-Mini-Instruct drops from 37.9 to 22.4 ). This strongly indicates that the selected SFT dataset is of poor quality or unsuitable for training these specific models. Because the proposed method (HD+IDFT) similarly fails to achieve substantial absolute gains—often improving by less than 1% over the original baseline (e.g., 49.6 to 49.8 for Qwen2.5-7B-instruct )—it is highly probable that the method simply excels at *avoiding* learning from bad data, rather than effectively learning from good data.

---

> ### Author Rebuttal · Authors · 2026-03-30
>
> # Response for reviewer GKNj
>
> ## W3 & Q1
>
> We agree with your insightful assessment. Following your suggestion, we re-conduct experiments using improved datasets and a broader range of models. **For the detailed results, please refer to the *'Additional experiments'* section in our response to Reviewer H8yp. we summarize the key information for you here:**
>
> - We use the recently released DeepMath dataset as the training set, where responses are generated by DeepSeek-R1 and verified against ground-truth answers, with substantially improved data quality.
> - Under the new experimental setting, we ensure that for every model, offline RL yields significant improvements over the original model, and our method consistently outperforms all offline RL baselines. These results are considerably more convincing than those reported previously.
> - On base models, SFT yields clear accuracy improvements. However, on instruct and thinking models, performance may still degrade, even when using distillation data. In recent years, instruct and reasoning models have typically already undergone distillation from larger models or RL prior to release, particularly on popular tasks such as mathematics. It is therefore entirely expected that open-source datasets fail to bring further improvements. This is also why prior works such as DFT and EAFT exclusively experiment on -base models rather than -instruct models. This is a limitation that cannot be resolved simply by changing datasets. This is precisely the problem that our proposed method, HD, is designed to address. We will incorporate this discussion into the Introduction as part of the background.
>
>
> ## W2 & Q3
>
> We clarify that our current scope focuses on tasks with well-defined answers, as stated in the paper (page 8, lines 397–398). In addition to mathematics, this setting naturally extends to domains such as code generation and agentic tasks, which already cover a broad range of applications.
>
> We agree that Hinted Decoding relies on ground-truth signals and therefore differs from reinforcement learning in exploratory settings. Our goal is not to replicate RL, but to provide a data-centric mechanism that improves training signals within supervised settings with available labeled data.
>
> As discussed in the paper (page 8, lines 394–397), our experiments do not indicate that the proposed approach can replace RLHF in preference learning or value alignment. **When we state that our work helps “bridge the gap”, we mean that it narrows the performance gap between SFT and RL in specific domains, rather than replacing RL in general.** We will further refine this wording to avoid potential misunderstandings.
>
>
> ## W1 & Q2
>
> To address this concern, we additionally include results with Online GRPO, as reported in the table in our response to reviewer 1Evf (Q 1&2). This provides a direct comparison with online RL settings.
>
> Our method operates at the dataset level and is therefore most directly comparable to offline RL methods, making such comparisons appropriate. **We further note that our method can be naturally extended to an online setting for direct comparison with online RL approaches, as discussed in the limitations (page 8, line 415).**
>
> ## Q4
>
> We clarify that $\tau$ is introduced in the ablation study to empirically validate the **Distribution Discriminant Theory** and to demonstrate the existence of an optimal learning interval (i.e., the *inverted-U curve*). The table below shows the average performance across the six mathematical benchmarks evaluated in our paper under different $\tau$ values.
>
>
> | Truncate $\tau$  | -1.0 | -2.0 | -3.0 | -4.0 | -5.0 | -6.0 | -7.0 | -8.0 | -9.0 | -10.0 |
> |-|-|-|-|-|-|-|-|-|-|-|
> | Qwen2.5-Math-7B | 36.67 | 36.49 | 36.82 | 37.41 | 37.23 | 37.85 | 37.93 | **38.08** | 37.23 | 37.57 |
> | Qwen2.5-Math-1.5B | 33.11 | 33.25 | 33.19 | 33.21 | 33.47 | 33.40 | **33.91** | 33.88 | 33.85 | 33.70 |
>
> As indicated by these empirical results, the performance of both models clearly follows an inverted-U trend. In the mathematical domain, the optimal truncation point consistently falls around **-7.0** to **-8.0** (peaking at 38.08 for *7B* and 33.91 for *1.5B*). In other domains, we can deduce that it will exhibit a slight negative shift, as the overall $\varphi$ tends to be smaller in novel domains.
>
> ## Summary
> We sincerely appreciate your constructive review. As discussed in our limitations and future work sections, we completely agree that there are many exciting avenues yet to be explored. We apologize if our phrasing caused misunderstanding that we aim to replace RL in all areas; we will revise the manuscript to clarify our scope. While a single paper cannot resolve every challenge, we hope that the theoretical contributions and broad applicability you generously highlighted as strengths offer meaningful value to the community. We view HD and IDFT as exploratory baselines that open promising new directions for future research.  Thank you for your time and effort!

---

> > ### Author Rebuttal · Reviewer_GKNj · 2026-04-03
> >
> > Based on the author's rebuttal, I increased my score.

---

### Official Review · Reviewer_1Evf · 2026-03-12

**Soundness:** 3
**Presentation:** 3
**Significance:** 3
**Originality:** 4
**Overall Recommendation:** 4
**Confidence:** 3

**Summary:**

This paper asks whether the on-policy benefits of RL can be transferred to SFT. To address this question, it proposes DDT, IDFT, and HD. The paper argues that, in domains with objective correctness, a main reason SFT fails is distribution mismatch. It then shows that simple corrections to both the loss and the data can bring SFT close to offline RL. The framework addresses three related problems: measuring whether a token or response lies within the current model distribution, modifying the SFT loss using that signal to reduce catastrophic forgetting, and rewriting the dataset so that correct answers are preserved while the style and presentation become closer to the model’s native distribution. More broadly, the paper suggests that RL generalizes better because it uses on-policy data and therefore stays closer to the model’s own distribution, whereas SFT may force the model to learn tokens that conflict with its existing knowledge structure.

**Compliance With Llm Reviewing Policy:**

Affirmed.

**Final Justification:**

Most of my concerns have been resolved, so I maintain a positive score.

**Key Questions For Authors:**

1. Recent off-policy RL methods, including off-policy GRPO, have shown performance close to on-policy methods. In that case, I wonder whether on-policy data is really the main reason RL generalizes better than SFT.
2. The paper discusses several offline RL methods, but it may also be useful to include a discussion of more recent approaches such as offline GRPO.

**Limitations:**

yes

**Strengths And Weaknesses:**

< Strengths >
1. The paper focuses on an important question: why RL generalizes better than SFT, and whether the on-policy property of RL can be transferred to SFT. This goes beyond changing a single loss and instead addresses a structural difference between SFT and RL.
2. The paper has good consistency across the theory, algorithm, and application sections.
3. The proposed methods, including IDFT, are logically well motivated.

< Weaknesses >
1. The mathematical expressions in Propositions 2 and 7 appear to be inconsistent with those in Appendix A.2.2.
2. Some claims in the paper read as somewhat stronger than what the empirical results support. For example, the quantitative results show that SimPO/Rej@4 performs better in some cases, while HD+IDFT appears competitive in others.

---

> ### Author Rebuttal · Authors · 2026-03-30
>
> # Response for reviewer 1Evf
>
> ## W1
>
> In our experiments, where the distributions $p$ and $q$ correspond to two GPT models, the magnitude of $(H(p_t) - H(q_t))$ is substantially smaller than $-D_{\mathrm{KL}}(q_t \| p_t)$, and thus the expression approximately equals $-D_{\mathrm{KL}}(q_t \| p_t)$. Furthermore, the experiments in Figure 2 empirically validate this proposition: the mean of data sampled from the model distribution is indeed centered around zero.
>
> However, we acknowledge that these observations do not constitute a rigorous justification. As a theoretical statement, for example when $q_t$ represents the distribution of a human-generated dataset and $H(q_t)$ is unknown and cannot be computed, this requires an explicit assumption. **We had originally planned to include additional experimental justification or formal assumptions; however, this was not fully incorporated in the submitted version, resulting in an imprecise statement of the proposition. We will revise the statements of Proposition 2 and 7 as well as Appendix A.2.2 to ensure consistency, and incorporate the entropy approximation as a clearly stated assumption in the revised version.**
>
>
> ## W2
> Thanks for the feedback. Following your suggestion and those of other reviewers, we re-conduct all experiments in Table 1 using an improved dataset. **Detailed results can be found in our response to reviewer H8yp (2. Additional Experiments). Here, we summarize the key takeaways:**
>
> - We use a newly released DeepMath as the training set. We include base, instruct, and thinking models, instead of just 3 different instruct models to enhance the generalization ability.
> - We refine the HD procedure by introducing an additional filtering step. After generating data via HD for the incorrectly answered samples in the training set, we introduce an additional filtering step: we filter out all samples that remain incorrect as well as those with $\bar{\varphi} < -0.05$ (i.e., samples that still deviate too far from the model distribution after HD, indicating they are too difficult).
> - In the original experiments, offline RL yielded less than a 1-point improvement over the base model, which limited the strength of the conclusions. **In the updated experiments, offline RL shows clear improvements. Moreover, our method achieves superior performance compared to Reject Sampling@16, DPO@16, and SimPO@16 across the evaluated models. For example, on Qwen2.5-7B (instruct), HD+IDFT achieves 36.21 / 53.50 on Math-C / Math-G, outperforming DPO@16 (34.49 / 52.41) and SimPO@16 (34.20 / 52.58).**
>
>
> ## Q 1&2
>
> We conduct additional experiments on GRPO using Qwen-2.5-7B-instruct with the open-r1 codebase and recommended hyperparameters.
>
>
> | **Method** | **(GPU Hours)** | **Math-C** | **Math-G** |
> |:---|:---:|:---:|:---:|
> | Origin | 0 | 33.48 | 52.32 |
> | DPO@16 | 197.6 | 34.49 | 52.41 |
> | SimPO@16 | 190.3 | 34.20 | 52.58 |
> | Offline GRPO@16 |  484.2  |  34.84  |  52.49  |
> | Online GRPO@16 (600 steps) |  385.35  |  34.50  |  52.37  |
> | Online GRPO@16 (1000 steps) |  662.4  |  37.78  |  **54.49**  |
> | Self-Distill | 115.7 | 35.09 | 52.39 |
> | **HD+SFT** | 139.6 | 36.63 | 53.42 |
> | **HD+IDFT** | 135.4 | 36.21 | 53.50 |
> | **HD+IDFT+Online GRPO** (600 steps) | 528.6 | **38.34** | 54.47 |
>
> We summarize the key observations as follows:
>
> - Offline GRPO does not substantially outperform other offline RL methods. Moreover, because we optimize the rollout procedure for the other offline RL baselines (see our response to reviewer H8yp (1. Description of experiments) regarding budget calculation), while GRPO must perform $K$ rollouts for every sample, its time cost is in fact higher.
> - Online GRPO shows comparable performance to other offline RL methods in the early stages of training. However, when training continues on the same dataset for a longer duration (e.g., 3–5 epochs), online GRPO continues to improve, whereas offline RL methods tend to plateau. This suggests that the open-source community may not always conduct experiments at sufficiently large scale due to limited computational resources, which could potentially lead to misleading conclusions.
> - We further observe that applying RL after HD+IDFT leads to improved performance, suggesting that the HD+IDFT stage enables the model to acquire new capabilities or features.
> - We further note that our method has the potential to be extended into an online variant in future work, which would enable a more direct comparison with online GRPO.
>
> It is still too early to draw firm conclusions about what RL provides beyond on-policy data. However, the role of on-policy data in mitigating catastrophic forgetting has been validated by a substantial body of prior work. From this perspective, HD has considerable potential for downstream task fine-tuning.
>
> Due to time constraints, experiments on additional models are still in progress, and we will include these results in the revised version.
>
> **Thanks for your valuable feedback and questions!**

---

> > ### Author Rebuttal · Reviewer_1Evf · 2026-04-02
> >
> > Most of my concerns have been resolved, so I maintain a positive score.

---

### Official Review · Reviewer_H8yP · 2026-03-14

**Soundness:** 4
**Presentation:** 4
**Significance:** 4
**Originality:** 3
**Overall Recommendation:** 5
**Confidence:** 3

**Summary:**

This paper first proposes DDT (Distribution Discriminant Theory). Based on the theory, IDFT (In-Distribution Finetuning) and HD (Hinted Decoding) are introduced: one enhances the generalization ability of SFT and the other provides an in-distribution generation paradigm while aligning with the ground truth. Rigorous theoretical proofs and empirical validation are provided for DDT. Experiments are conducted in two main settings: comparison with enhanced SFT works and comparison with offline RL. Results show that IDFT consistently outperforms baselines in the first setting, and HD+IDFT achieves comparable performance in the second setting.

**Compliance With Llm Reviewing Policy:**

Affirmed.

**Final Justification:**

My main concern was the clarity of the HD+SFT/IDFT experimental procedure. The rebuttal addressed this fully, detailing the rollout strategy, filtering procedure, and budget computation. The additional experiments — covering base, instruct, and thinking models with a stronger dataset and self-distillation baseline — further strengthen the results.

The rebuttal resolved my concerns and reinforced my prior positive assessment. I maintain my recommendation to accept.

**Key Questions For Authors:**

In the SFT vs Offline RL setting, how is the HD combined with SFT/IDFT? Specifically: (a) How many rollouts are used, and (b) How is the training data selected?

A clearer description of this procedure would help assess the practical reproducibility.

**Limitations:**

yes

**Strengths And Weaknesses:**

- Soundness

  The submission is technically sound. The DDT theory is supported by rigorous mathematical proofs and empirical validation. The IDFT algorithm naturally follows from the theoretical framework, with gradient analysis supporting its effectiveness. Hinted Decoding (HD) combines imitation decoding and original decoding through entropy-based weighting, which is well motivated by DDT. While I have not checked every proof, the assumptions are reasonable. The experiments are well-designed, and the benchmarks cover both mathematical reasoning and general reasoning tasks. The set of baselines is comprehensive, and the evaluation includes everything I wanted to see.

  *Minor Note: There is a typographical error in Table 1 — General-R is shown as General-G in the caption.*

- Presentation

  The paper is clearly written and well structured. The overall narrative is easy to follow. The work properly positions itself in the context of prior and concurrent literature and clearly discusses how it differs.

- Significance

  The paper makes a significant theoretical contribution by presenting the Distribution Discriminant Theory. The experimental results show a significant improvement to the capability of SFT, achieving state-of-the-art performance among enhanced SFT works. The comparison between HD+SFT and SFT/Rej@2 shows that SFT benefits from in-distribution and correct data generated by HD. I believe both IDFT and HD can have a broad impact on LLM finetuning and future work.

- Originality

  The proposed DDT theory, IDFT, and HD are new to me and differ from existing methods. The novelty is well justified.

---

> ### Author Rebuttal · Authors · 2026-03-30
>
> # Response for reviewer H8yp
>
> We sincerely appreciate your recognition of the significance and novelty of our work! We will correct the typo in Table 1 and revise the description issues as suggested.
>
>  ## 1. Description of experiments
>
> - **Offline RL Rollouts**: For the @K setting in offline RL, we adopt a highly cost-efficient procedure for each method. Taking DPO@16 as an example, for each problem, we treat a sample with a correct answer as the selected response and a sample with an incorrect answer as the rejected response. Rather than directly rolling out 16 times per problem (which would increase the cost of DPO while favoring its performance), we first roll out 2 samples per problem, filter out all problems that happen to have exactly one correct and one incorrect response, and then proceed with a third rollout for the remaining problems, repeating this process until each problem has at most 16 rollouts. This strategy enhances the efficiency of the baseline comparison.
> - **HD Rollouts**: For HD, we first perform one rollout per problem and then apply HD only to incorrectly answered samples. During the HD process, we use $\beta=10$ and perform a single rollout for all incorrect samples. Subsequently, we filter out all samples that remain incorrect as well as those with $\bar{\varphi} < -0.05$ (i.e., samples that still deviate too far from the model distribution after HD, indicating they are too difficult). The final training data is composed of the successfully corrected samples from HD together with the samples that the model answered correctly in the initial rollout.
> - **Budget computation**: For all rollouts, we leverage the vLLM package for accelerated inference, and report the total GPU time encompassing both generation and training.
>
> We will incorporate these details into the revised version. We sincerely appreciate the reviewer’s valuable suggestions and thank you again for your recognition of our work.
>
> ## 2. Additional experiments
>
> Following the suggestions of other reviewers, we re-conduct the experiments of Table 1 under improved settings. We also include the main results here for completeness. The key differences are as follows:
>
> - We include base, instruct, and thinking models, instead of just three different instruct models to enhance the generalization ability.
> - We implement the CUDA kernel for the vLLM support of Hinted Decoding. We then change the unit of computational budget to overall GPU Hours instead of FLOPs (The generation process is memory-bounded, and therefore FLOPs can significantly underestimate the actual latency.).
> - We adopt a higher-quality training dataset (DeepMath-103K). This ensures that offline RL consistently yields significant improvements over the baseline, thereby strengthening the reliability of our experimental results.
> - We additionally include a self-distillation baseline (imitation baseline).
>
> The new result table which shows a much larger gain than previous one is as follows:
>
> | Qwen2.5 | -7B  | (base) |  |  | Qwen2.5-7B  | -instruct | (instruct) |  | DeepSeek-R1  | -distill-Qwen-7B | (thinking) |  |
> |:---|:---:|:---:|:---:|:---:|:---:|:---:|:---:|:---:|:---:|:---:|:---:|:---:|
> | **Method** | **(GPU Hour)** | **Math-C** | **Math-G** | **General-R** | **(GPU Hour)** | **Math-C** | **Math-G** | **General-R** | **(GPU Hour)** | **Math-C** | **Math-G** | **General-R** |
> | Origin | 0 | 22.06 | 42.13 | 48.28 | 0 | 33.48 | 52.32 | 61.80 | 0 | 43.59 | 54.68 | 59.38 |
> | SFT | 32.85 | 23.93 | 44.24 | 52.10 | 33.25 | 30.07 | 48.33 | 59.93 | 33.04 | 39.43 | 53.52 | 58.28 |
> | DFT | 32.84 | 17.09 | 31.34 | 51.93 | 34.01 | 30.22 | 48.16 | 56.08 | 33.49 | 39.52 | 53.03 | 57.39 |
> | EAFT | 33.77 | 23.95 | 42.88 | 53.57 | 33.79 | 30.98 | 48.40 | 60.21 | 33.98 | 39.49 | 52.93 | 59.61 |
> | Rej@16 | 221.9 | 23.62 | 42.89 | 47.43 | 191.2 | 34.19 | 53.17 | 60.43 | 869.6 | 42.68 | 54.46 | 59.42 |
> | DPO@16 | 276.4 | 22.43 | **45.95** | 48.29 | 197.6 | 34.49 | 52.41 | 59.84 | 821.2 | 43.35 | 55.18 | 59.33 |
> | SimPO@16 | 230.1 | 20.23 | 38.45 | 47.93 | 190.3 | 34.20 | 52.58 | 60.30 | 801.5 | 43.80 | 54.43 | 59.27 |
> | Self-Distill | 324.7 | 24.96 | 44.07 | 52.99 | 115.7 | 35.09 | 52.39 | 60.25 | 249.3 | 43.74 | **54.90** | 59.79 |
> | **HD+SFT** | 212.1 | **25.54** | 45.14 | 52.94 | 139.6 | **36.63** | **53.42** | 60.30 | 424.9 | **44.63** | 54.84 | 60.16 |
> | **HD+IDFT** | 214.8 | **27.37** | **47.07** | 53.34 | 135.4 | **36.21** | **53.50** | 60.41 | 425.5 | **43.83** | **55.51** | 59.95 |

---

> > ### Author Rebuttal · Reviewer_H8yP · 2026-04-03
> >
> > Thank you for the thorough rebuttal. I keep my positive score.

---

### Decision · Program_Chairs · 2026-04-30

**Decision:**

Accept (regular)

**Comment:**

This paper addresses the generalization gap between SFT and RL in LLMs. The authors posit that standard SFT uniformly fits external (potentially out-of-distribution) data, which can damage a model's pre-trained knowledge. To bridge this gap, the paper introduces an on-policy SFT framework. The framework implements two mechanisms: in-distribution finetuning to dynamically reweight the SFT loss (suppressing out-of-distribution noise), and hinted decoding to rewrite training data into a style native to the model. Experiments on math and coding tasks demonstrate that this approach achieves generalization comparable to offline RL algorithms (like DPO) while maintaining the computational efficiency of standard SFT.

Strengths:
- The paper presents a sound mathematical framework, analyzing why RL generalizes better than standard SFT.
- Bridging SFT and RL through data-level and loss-level interventions is a compelling and well-executed approach.
- Achieving RL-level generalization with SFT-level computational costs is a highly valuable practical contribution to the field.

Weaknesses:
- The original submission lacked comparisons against more modern online and off-policy RL techniques, such as GRPO.
-There were questions regarding whether the perceived performance gains were artificially inflated by the quality of the original baseline datasets.
- The initial manuscript lacked sufficient detail regarding the experimental procedures for rollouts, data selection, and mathematical notations.

Rebuttal:
- The authors ran new experiments using a higher-quality dataset (DeepMath).
- The authors included the requested online and offline GRPO baselines, demonstrating that their method remains competitive against state-of-the-art RL approaches.
- The authors documented their data filtering strategies and rollout budgets, and corrected the minor mathematical notation inconsistencies pointed by reviewers.

Two reviewers recommended acceptance. While one reviewer maintained a weak reject rating (upgraded from a lower initial rating), the integration of the DeepMath dataset and the addition of GRPO baselines should have addressed their concerns. Overall, this is a weak accept.